# Personalized Feature Translation for Expression Recognition: An Efficient Source-Free Domain Adaptation Method

**Masoumeh Sharafi**[1], **Soufiane Belharbi**[1], **Muhammad Osama Zeeshan**[1],
**Houssem Ben Salem**[1], **Ali Etemad**[5], **Alessandro Lameiras Koerich**[2],
**Marco Pedersoli**[1], **Simon L Bacon**[3,4] & **Eric Granger**[1]

[1]LIVIA, Dept. of Systems Engineering, ETS Montreal, Canada
[2]LIVIA, Dept. of Software and IT Engineering, ETS Montreal, Canada
[3]Dept. of Health, Kinesiology, & Applied Physiology, Concordia University, Montreal, Canada
[4]Montreal Behavioural Medicine Centre, CIUSSS Nord-de-l'Ile-de-Montréal, Canada
[5]Dept. of Electrical and Computer Engineering, Queen's University, Kingston, Canada

`{masoumeh.sharafi.1, muhammad-osama.zeeshan.1, houssem.ben-salem.1}@ens.etsmtl.ca`
`{soufiane.belharbi, marco.pedersoli, alessandro.koerich, eric.granger}@etsmtl.ca`
`ali.etemad@queensu.ca, simon.bacon@concordia.ca`

## Abstract

Facial expression recognition (FER) models are employed in many video-based affective computing applications, such as human-computer interaction and healthcare monitoring. However, deep FER models often struggle with subtle expressions and high inter-subject variability, limiting their performance in real-world applications. To improve performance, source-free domain adaptation (SFDA) methods have been proposed to personalize a pretrained source model using only unlabeled target domain data, thereby avoiding data privacy, storage, and transmission constraints. This paper addresses a common challenging scenario where source data is unavailable for adaptation, and only unlabeled target data consisting solely of neutral expressions is available. SFDA methods are not typically designed to adapt using target data from only a single class. Further, using models to generate facial images with non-neutral expressions can be unstable and computationally intensive. In this paper, the Source-Free Domain Adaptation with Personalized Feature Translation (`SFDA-PFT`) method is proposed for SFDA. Unlike current image translation methods for SFDA, our lightweight method operates in the latent space. We first pre-train the translator on source domain data to transform the subject-specific style features from one source subject into another. Expression information is preserved by optimizing a combination of expression consistency and style-aware objectives. Then, the translator is adapted to neutral target data, without using source data or image synthesis. By translating in the latent space, `SFDA-PFT` avoids the complexity and noise of face expression generation, producing discriminative embeddings optimized for classification. Using `SFDA-PFT` eliminates the need for image synthesis, reduces computational overhead, and only adapts a lightweight translator, making the method efficient compared to image-based translation. Our extensive experiments on four challenging video FER benchmark datasets, `BioVid`, `StressID`, `BAH`, and `Aff-Wild2`, show that `SFDA-PFT` consistently outperforms state-of-the-art SFDA methods, providing a cost-effective approach that is suitable for real-world, privacy-sensitive FER applications.
Our code is publicly available at: github.com/MasoumehSharafi/SFDA-PFT.

## 1 Introduction

FER plays an important role in video-based affective computing, enabling systems to interpret the emotional or health states of humans through non-verbal cues (Calvo & D'Mello, 2010; Ko,

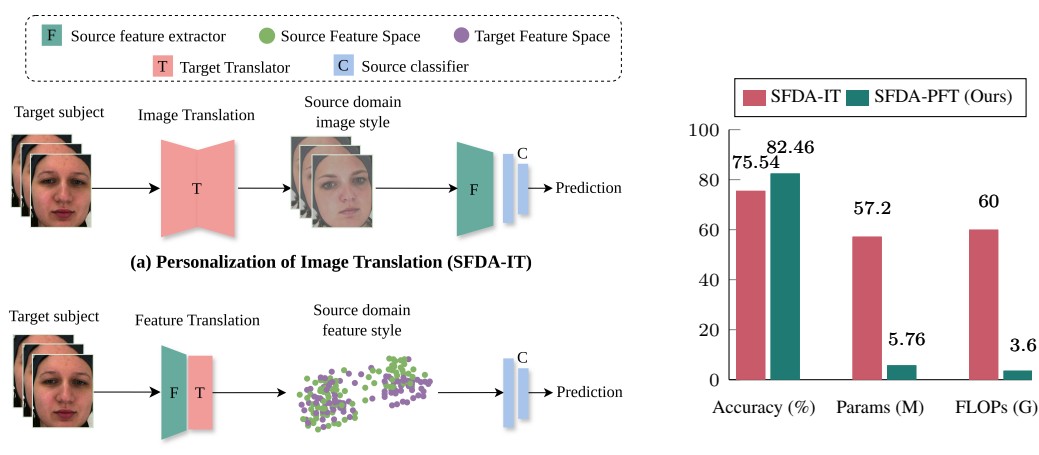

Figure 1: A comparison between standard image translation, SFDA-IT (Hou & Zheng, 2021a), against our `SFDA-PFT` on `BioVid` data. (a) Image translation methods operate at the pixel level and require complex mappings to align the target and source styles. (b) Our `SFDA-PFT` method directly translates in the source feature space, allowing for efficient personalization. (*right*) Accuracy, parameter counts, and FLOPs at inference highlight the trade-offs between the two approaches, with models implemented using a `ResNet-18` backbone.

2018). Its applications range from human-computer interaction (Pu & Nie, 2023), to health monitoring (Gaya-Morey et al., 2025), and clinical assessment of pain, depression and stress (Calvo & D'Mello, 2010). Despite recent advances in deep learning (Barros et al., 2019; Sharafi et al., 2022; 2023) and the availability of large annotated datasets for training (Walter et al., 2013; Kollias & Zafeiriou, 2019), deep FER models may perform poorly when deployed on data from new users and operational environments. This is due to the mismatch between distributions of the training (source domain) data and the testing (target operational domain) data. Beyond variations in capture conditions, data distributions may differ significantly across subjects. Inter-subject variability (Zeng et al., 2018; Martinez, 2003) can degrade the accuracy and robustness of deep FER models in real-world applications (Li & Deng, 2020a; Zhao et al., 2016).

To improve performance, this paper focuses on subject-based adaptation or personalization of deep FER models to video data from target subjects. Various unsupervised domain adaptation (UDA) methods have been proposed to address the distribution shifts by aligning feature distributions (Li & Deng, 2018; Zhu et al., 2016; Chen et al., 2021; Li & Deng, 2020b). However, they typically require access to labeled source data during adaptation, a constraint that is often infeasible in privacy-sensitive application areas like healthcare due to concerns for data privacy, data storage, and computation costs. This has led to the emergence of source-free domain adaptation (SFDA), where adaptation of a pretrained source model is performed using only unlabeled target data (Liang et al., 2020; Tang et al., 2024; Guichemerre et al., 2024). These methods (Fang et al., 2024; Li et al., 2024) can be broadly categorized into (1) model-based approaches, which adapt the model parameters, using target domain statistics or pseudo-labels, and (2) data-based approaches (focus of this paper), which instead operate at the data level by translating target images into the source domain style, enabling inference through the frozen source model without modifying its parameters.

State-of-the-art SFDA methods assume access to data from all target classes, which is not practical in real-world FER applications. Indeed, person-specific data representing non-neutral expressions is typically costly or unavailable. A short neutral control video may, however, be collected for target individuals and used to personalize a model to the variability of an individual's diverse expressions. In practice, collecting and annotating neutral target data for adaptation is generally easier and less subjective than gathering non-neutral emotional data. Recent work has employed GANs to generate expressions based on neutral inputs but relies on image-level disentanglement of identity and expression, which is often unstable and computationally expensive (Sharafi et al., 2025). This limitation reduces the effectiveness of model-based adaptation or fine-tuning strategies, particularly those relying on pseudo-labeling, since labels for neutral data are available during adaptation. However,

data-based strategies translate target data into the source domain style. This avoids adapting parameters of the source classifier and enables direct inference with the frozen source model, improving stability, efficiency, and privacy. Following this direction, some SFDA methods (e.g., SFDA-IT) leverage generative models to translate target inputs into source-style images, guided by the source model (Hou & Zheng, 2021a;b). However, these methods are not adapted for subject-specific adaptation of FER models. They consider the source as a single domain and often suppress important subject-specific cues for personalized FER. They also depend on expressive target data, which is rarely available in practice, making generative training infeasible in limited-data settings.

To address the limitations of image translation methods for SFDA, we introduce the Source-Free Domain Adaptation with Personalized Feature Translation (`SFDA-PFT`) method that explicitly models subject-specific variation within the source domain. `SFDA-PFT` is a conceptually simple yet effective feature translation method for source-free personalization in FER. The key idea is to pre-train a translator network that maps features from one source subject to another while preserving the underlying expression. This subject-swapping objective encourages the model to capture intra-class, inter-subject variability, learning the structural relationship between expression and identity-specific features within the source domain. During adaptation, only a small subset of the translator's parameters is fine-tuned to translate the style of the target subject, enabling stable and cost-effective personalization. Figure 1 (left) illustrates the difference between image-based and feature-based translation. Image-level methods (Figure 1(a)) generate target images in the source style, relying on complex generative models that introduce instability and high computational overhead. In contrast, Figure 1(b) shows that `SFDA-PFT` translates target features directly toward the closest source subject, preserving expression without pixel-level synthesis. The complexity comparison in Figure 1 (right) shows that `SFDA-PFT` achieves higher accuracy while requiring up to $100\times$ fewer parameters and $17\times$ fewer FLOPs than SFDA-IT, highlighting its efficiency and suitability for deployment.

**Our contributions. (1)** We propose a personalized feature translation (`SFDA-PFT`) method for SFDA in FER using only target images with neutral expressions. Unlike image translation methods that require expressive target data and generative models, our approach translates features across subjects while preserving expression semantics. Adaptation is performed in the feature space with only a small subset of parameters and a significant reduction in computational complexity. **(2)** Style-aware and expression consistency losses are proposed to guide the translation process without requiring expressive target data. Our method only requires a few neutral target samples for lightweight adaptation, introduces no additional parameters at inference time, and ensures stable and cost-effective deployment. **(3)** An extensive set of experiments is provided on four video FER benchmarks, `BioVid` (pain estimation), `StressID` (stress recognition), `BAH` (ambivalence-hesitancy recognition), and `Aff-Wild2` (basic expression classification). Results show that our `SFDA-PFT` achieves performance that is comparable to or higher than state-of-the-art SFDA (pseudo-labeling and image translation) methods, with lower computational complexity.

## 2 Related Work

### 2.1 Facial Expression Recognition

FER aims to identify human emotional states from facial images or video sequences. To enhance generalization, UDA methods (Feng et al., 2023; Cao et al., 2018; Chen et al., 2021; Ji et al., 2019; Li & Deng, 2020b) and multi-source domain adaptation (MSDA) techniques (Zhou et al., 2024) align distributions between source and target domains using unlabeled target data. While effective, these approaches typically require access to source data during adaptation. Personalized FER methods (Yao et al., 2021; Kollias et al., 2020) adapt models to individual users but rely on labeled data per user. More recent subject-aware adaptation frameworks (Zeeshan et al., 2024; 2025b;a) treat each subject as a domain and adapt across users, yet still depend on source data. These constraints motivate the need for SFDA, which enables model personalization without accessing source samples, offering a more practical solution for privacy-sensitive FER applications.

### 2.2 Source-Free Domain Adaptation and Personalization

SFDA addresses privacy, computational and storage concerns by adapting a pre-trained source model to an unlabeled target domain without access to source data. Common model-based strate-

gies include self-supervised learning (Yang et al., 2021; Litrico et al., 2023), pseudo-labeling (Liang et al., 2020), entropy minimization (Liang et al., 2020), and feature alignment via normalization or auxiliary modules (Li et al., 2016; Liang et al., 2022; Kim et al., 2021b). SHOT (Liang et al., 2020) and DINE (Liang et al., 2022) exemplify efficient adaptation via classifier tuning or Batch-Norm statistics. However, these methods often assume confident predictions and smooth domain shifts, which are frequently violated in FER due to high inter-subject variability and subtle expression differences. FER-specific adaptations such as CluP (Conti et al., 2022) and FAL (Zheng et al., 2025) address label noise and pseudo-label instability, yet challenges remain when only neutral target expressions are available. DSFDA (Sharafi et al., 2025) tackles this by disentangling identity and expression using generative models, but its reliance on adversarial training and multi-stage pipelines limits scalability and robustness in practical deployment.

## 2.3 FEATURE TRANSLATION FOR SFDA

Image translation is a common data-based strategy for SFDA that maps target images into the source style using generative models, allowing frozen source models to generalize without access to source data (Hou & Zheng, 2021a;b; Kurmi et al., 2021; Qiu et al., 2021; Tian et al., 2021; Ding et al., 2022). While effective in general tasks, these methods face critical limitations in FER and personalization. FER requires preserving subtle expression cues and identity-specific features, which are often distorted by image synthesis. Moreover, generative models are computationally intensive, prone to instability, and assume access to expressive target samples, an unrealistic assumption in neutral-only personalization settings. To address these challenges, we propose translating features instead of images, using a compact, self-supervised translator that maps target features into the source-aligned space without requiring adversarial training, source data, or expressive target inputs, offering a stable and efficient solution for source-free FER personalization.

## 3 PROPOSED PERSONALIZED FEATURE TRANSLATION METHOD

Figure 2 illustrates the overall framework of our `SFDA-PFT` method. The source model is comprised of a feature extractor backbone and classifier head, both frozen during adaptation. To adapt this model to a new target subject with only a few neutral images extracted from a video, we introduce a translator network, a copy of the source encoder equipped with lightweight adaptation layers after the feature extractor. The translator is pretrained on source data using a subject-swapping

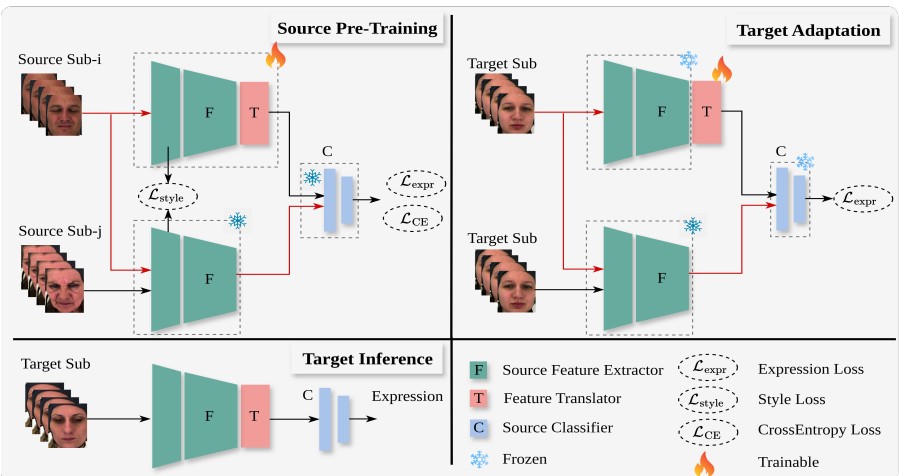

Figure 2: Overview of the proposed `SFDA-PFT` method. (a) During pre-training, the translator **T** is trained to map Sub-i features into the distribution of Sub-j from the source dataset, using a combination of style alignment and expression consistency losses. (b) During adaptation, only the feature translator **T** is updated using expression-consistent predictions from two different images (Image1 and Image2) of the same target subject. (c) At inference time, the trained translator **T** and the fixed source classifier **C** are used to predict expressions for target-domain inputs.

objective: translating features between source subjects while maintaining expression labels. This enables the model to capture subject-specific information and preserve expression.

**Architecture:** Let $\mathcal{D}_S = \{(\mathbf{x}_s, y_s)\}$ be a labeled source dataset, where $\mathbf{x}_s$ is a source subject and $y_s \in \mathcal{Y}$ its corresponding expression label. Let $\mathcal{D}_T = \{\mathbf{x}_t\}$ denote the unlabeled dataset for a target subject. We denote by $\mathbf{F}$ the source feature extractor and by $\mathbf{C}$ the classifier head. The translator network is defined as the composition of $\mathbf{F}$ followed by a set of lightweight, subject-adaptive layers $\mathbf{T}$. Thus, the translator $\mathbf{T}_{\text{full}} = \mathbf{T} \circ \mathbf{F}$ takes an image as input and outputs a translated feature representation. The source classifier $(\mathbf{F}, \mathbf{C})$ is trained on $\mathcal{D}_S$ and remains frozen during adaptation. The translator is first pretrained on $\mathcal{D}_S$ to learn identity transformation while preserving expression, and then adapted to each target subject individually using only a few samples.

### 3.1 SOURCE PRE-TRAINING

The objective of source pre-training is twofold: first, to train a reliable expression classifier on labeled source data, and second, to initialize the translator network so that it can disentangle and recompose identity and expression in the feature space. This initialization is crucial because the translator will later be adapted to new subjects using only a few unlabeled samples.

Formally, the source classifier consists of a feature extractor $\mathbf{F}$ and a classifier head $\mathbf{C}$, which are optimized on the source dataset $\mathcal{D}_S = \{(\mathbf{x}_s, y_s)\}$ by minimizing the standard cross-entropy loss:

$$\mathcal{L}_{\text{CE}}(\mathbf{x}_s, y_s) = -\log \big[\, \mathbf{C}(\mathbf{F}(\mathbf{x}_s)) \,\big]_{y_s}. \tag{1}$$

To pre-train the translator $\mathbf{T}$, we construct pairs of source images $(\mathbf{x}_i, \mathbf{x}_j)$ from distinct subjects. The first image $\mathbf{x}_1$ carries the expression that should be preserved, while the second $\mathbf{x}_j$ provides the target identity to which the representation should adapt. Extracted features are denoted as $\mathbf{f}_i = \mathbf{F}(\mathbf{x}_i), \mathbf{f}_j = \mathbf{F}(\mathbf{x}_j), \hat{\mathbf{f}}_i = \mathbf{T}(\mathbf{f}_i)$. The translated representation $\hat{\mathbf{f}}_i$ is optimized with two complementary criteria. First, expression semantics are preserved by minimizing the divergence between classifier predictions on the original and translated features:

$$\mathcal{L}_{\text{expr}} = D_{\text{KL}}\Big( \mathbf{C}(\mathbf{f}_i) \,\|\, \mathbf{C}(\hat{\mathbf{f}}_i) \Big). \tag{2}$$

Second, the translated feature is explicitly encouraged to adopt the identity statistics of the reference subject $\mathbf{x}_j$. Rather than relying on pixel-level synthesis or adversarial identity matching, we achieve this alignment directly in feature space by matching low-order statistics of early-layer activations. Concretely, for each selected layer $l \in \mathcal{L}$, we compute the per-channel mean $\mu(\cdot)$ and standard deviation $\sigma(\cdot)$ of both the translated representation $\hat{\mathbf{f}}_i^l$ and the reference identity feature $\mathbf{f}_j^l$, and minimize their squared differences. The resulting objective:

$$\mathcal{L}_{\text{style}} = \sum_{l \in \mathcal{L}} \Big( \|\mu(\hat{\mathbf{f}}_i^l) - \mu(\mathbf{f}_j^l)\|_2^2 + \|\sigma(\hat{\mathbf{f}}_i^l) - \sigma(\mathbf{f}_j^l)\|_2^2 \Big), \tag{3}$$

forces the translator to reshape the distribution of $\hat{\mathbf{f}}_1$ so that it reflects the identity-specific style of $\mathbf{x}_2$ while leaving expression semantics intact.

This formulation is inspired by the observation that per-channel statistics encode subject-dependent appearance cues (e.g., facial geometry, texture, or lighting) that are orthogonal to expression dynamics. By matching only the first two moments, the translator adapts identity without overfitting to sample-specific details, thus avoiding artifacts that commonly arise in image-level translation. Crucially, this lightweight alignment in feature space is both efficient and robust to noise, making it a key ingredient of our method. The final source pre-training objective combines these components:

$$\mathcal{L}_{\text{source}} = \mathcal{L}_{\text{CE}} + \lambda_{\text{expr}} \mathcal{L}_{\text{expr}} + \lambda_{\text{style}} \mathcal{L}_{\text{style}}. \tag{4}$$

where $\lambda_{\text{expr}}$ and $\lambda_{\text{style}}$ weight the preserving expression semantics and aligning subject identity.

### 3.2 TARGET ADAPTATION AND INFERENCE

Given a small set of unlabeled frames from a new target subject, the goal is to personalize the translator $\mathbf{T}_{\text{full}}$ while keeping the source classifier $(\mathbf{F}, \mathbf{C})$ fixed. Adaptation is performed independently

for each subject and updates only the lightweight adaptive layers $\mathbf{T}$, ensuring efficiency and avoiding catastrophic interference with previously learned knowledge. Since all target samples originate from the same identity, explicit identity alignment is unnecessary; the adaptation stage thus focuses exclusively on preserving expression semantics. For each target frame $\mathbf{x}_t$, features are first extracted by the frozen source encoder as $\mathbf{f}_t = \mathbf{F}(\mathbf{x}_t)$ and then transformed by the translator as $\hat{\mathbf{f}}_t = \mathbf{T}(\mathbf{f}_t)$.

To maintain expression fidelity, we enforce consistency between classifier predictions before and after translation by minimizing the KL divergence:

$$\mathcal{L}_{\text{expr}} = D_{\text{KL}}\Big(\mathbf{C}(\mathbf{f}_t) \,\|\, \mathbf{C}(\hat{\mathbf{f}}_t)\Big). \tag{5}$$

This self-distillation objective anchors the adapted translator to the original classifier's decision boundary, ensuring that the expression information present in $\mathbf{f}_t$ is preserved after subject-specific transformation. Since labels are not required, even a few neutral frames are sufficient for adaptation. In practice, this enables efficient and data-light personalization that can be performed at test time without revisiting the source dataset.

**Inference.** After adaptation, the personalized translator $\mathbf{T}_{\text{full}} = \mathbf{T} \circ \mathbf{F}$ is used for recognition. For a new frame $\mathbf{x}_t$ of the same target subject, the translator maps its features into a source-aligned representation while maintaining the expression. The frozen classifier $\mathbf{C}$ then predicts the expression from adapted features. This design allows test-time subject personalization without labels, avoids storing or accessing source data during deployment, and eliminates the overhead of pixel-level translation. As a result, the method provides a lightweight yet effective strategy for SFDA in FER, combining the stability of frozen discriminative models with the flexibility of subject-adaptive translation.

## 4 RESULTS AND DISCUSSION

### 4.1 EXPERIMENTAL METHODOLOGY

**Datasets:** In our experiments, we evaluate on four diverse facial expression datasets: `BioVid` (Walter et al., 2013), which contains controlled laboratory recordings of pain stimuli; `StressID` (Chaptoukaev et al., 2023), which captures stress levels based on self-reports; `BAH` (González-González et al., 2025), a large-scale dataset for recognizing ambivalence and hesitancy expressions in naturalistic recordings; and `Aff-Wild2` (Kollias & Zafeiriou, 2019), a widely used in-the-wild benchmark for basic expression recognition. These datasets collectively cover a range of domains, from controlled lab settings to real-world scenarios, and from binary (pain, stress, ambivalence/hesitancy) to multi-class (seven basic emotions) classification tasks. Full dataset descriptions are provided in the Appendix.

**Protocol:** In experiments, each subject is viewed as an independent target domain. In the `BioVid`, `BAH`, `Aff-Wild2`, and `StressID` datasets. Following prior works on personalization of FER (Zeeshan et al., 2024; 2025b; Sharafi et al., 2025), we evaluate on the standard 10 fixed target subjects used in the literature, which span both genders and cover a range of ages, with each subject contributing hundreds to thousands of frames. This protocol enables fair comparison while ensuring that per-subject metrics are computed over large sample sizes. This subject-specific setup reflects real-world personalization scenarios and enables assessment under inter-subject variability. During adaptation, we assume access only to neutral expression data from the target subjects. No source data are available at this stage, consistent with the SFDA setting. We evaluate performance under the following four settings. **Source-Only.** The model is trained on labeled source-domain data and directly evaluated on target subjects without adaptation. This serves as a lower-bound baseline, highlighting the impact of domain shift. **SFDA (model-based).** The model is adapted using only neutral data from the target domain. We compare our proposed `SFDA-PFT` method with recent state-of-the-art SFDA methods, including SHOT (Liang et al., 2020), TPDS (Tang et al., 2024), NRC (Yang et al., 2021), SFIT (Hou & Zheng, 2021b), SFDA-IT (Hou & Zheng, 2021a), and DSFDA (Sharafi et al., 2025). **SFDA (data-based).** This variant incorporates our subject-specific translation module, which aligns target features to the source domain through subject-specific adaptation. **Oracle.** The model is fine-tuned using labeled target-domain data, including neutral and non-neutral expressions.

Table 1: Comparison of our proposed `SFDA-PFT` method against state-of-the-art SFDA methods on the `BioVid` dataset (10 target subjects, 77 source subjects). Bold numbers indicate the best `F1`.

| Setting | Methods | Sub-1 | Sub-2 | Sub-3 | Sub-4 | Sub-5 | Sub-6 | Sub-7 | Sub-8 | Sub-9 | Sub-10 | Average |
|---|---|---|---|---|---|---|---|---|---|---|---|---|
| Source-only | Source model (no adaptation) | 62.78 | 52.76 | 82.02 | 80.83 | 82.73 | 56.03 | 71.85 | 66.90 | 50.01 | 45.79 | 65.17 |
| SFDA (model-based) | SHOT (Liang et al., 2020) | 52.97 | 45.35 | 38.98 | 49.80 | 51.92 | 46.43 | 51.72 | 46.74 | 52.10 | 42.20 | 47.82 |
| | NRC (Yang et al., 2021) | 48.45 | 32.16 | 68.60 | 59.52 | 65.06 | 34.85 | 52.20 | 44.06 | 44.82 | 34.68 | 48.44 |
| | TPDS (Tang et al., 2024) | 62.26 | 53.16 | 75.23 | 64.79 | 87.06 | 56.14 | 58.20 | 65.84 | 54.24 | 45.79 | 62.27 |
| | DSFDA (Sharafi et al., 2025) | 65.72 | 64.10 | 77.57 | 73.12 | 75.20 | 57.59 | 76.15 | 74.73 | 59.08 | 61.54 | 68.48 |
| SFDA (data-based) | SFIT (Hou & Zheng, 2021b) | 76.85 | 65.33 | 78.70 | 80.44 | 87.01 | 54.44 | 57.54 | 70.81 | 57.66 | **75.92** | 70.47 |
| | SFDA-IT (Hou & Zheng, 2021a) | 71.54 | 63.89 | 84.53 | 80.30 | 86.24 | 59.18 | 77.66 | 72.08 | 54.97 | 67.01 | 71.74 |
| | **SFDA-PFT (ours)** | **80.65** | **71.75** | **90.26** | **81.54** | **92.68** | **70.06** | **84.26** | **79.29** | **74.53** | 58.08 | **78.31** |
| Oracle | Supervised fine-tuning | 92.22 | 86.83 | 91.89 | 92.96 | 91.27 | 87.65 | 85.48 | 90.30 | 93.28 | 92.12 | 90.40 |

Table 2: Comparison of our proposed `SFDA-PFT` method against state-of-the-art SFDA methods on the `StressID` dataset (10 target subjects, 44 source subjects). Bold numbers indicate the best `F1`.

| Setting | Methods | Sub-1 | Sub-2 | Sub-3 | Sub-4 | Sub-5 | Sub-6 | Sub-7 | Sub-8 | Sub-9 | Sub-10 | Average |
|---|---|---|---|---|---|---|---|---|---|---|---|---|
| Source-only | Source model (no adaptation) | 44.44 | 43.54 | 45.34 | 44.89 | 45.79 | 43.99 | 45.34 | 44.89 | 44.44 | 45.34 | 44.80 |
| SFDA (model-based) | SHOT (Liang et al., 2020) | 42.66 | 41.79 | 43.52 | 43.09 | 43.95 | 42.22 | 43.52 | 43.09 | 42.66 | 43.52 | 43.00 |
| | NRC (Yang et al., 2021) | 40.67 | 39.85 | 41.49 | 41.08 | 41.90 | 40.26 | 41.49 | 41.08 | 40.67 | 41.49 | 41.00 |
| | TPDS (Tang et al., 2024) | 50.10 | 49.08 | 51.11 | 50.60 | 51.61 | 49.59 | 51.11 | 50.60 | 50.10 | 51.11 | 50.50 |
| | DSFDA (Sharafi et al., 2025) | 65.47 | 64.15 | 66.79 | 66.13 | 67.45 | 64.81 | 66.79 | 66.13 | 65.47 | 66.79 | 66.00 |
| SFDA (data-based) | SFIT (Hou & Zheng, 2021b) | 62.00 | 60.75 | 63.25 | 62.63 | 63.88 | 61.37 | 63.25 | 62.63 | 62.00 | 63.25 | 62.50 |
| | SFDA-IT (Hou & Zheng, 2021a) | 63.19 | 61.91 | 64.47 | 63.83 | 65.10 | 62.55 | 64.47 | 63.83 | 63.19 | 64.47 | 63.70 |
| | **SFDA-PFT (ours)** | **69.36** | **67.96** | **70.76** | **70.06** | **71.46** | **68.66** | **70.76** | **70.06** | **69.36** | **70.76** | **69.92** |
| Oracle | Supervised fine-tuning | 96.72 | 94.76 | 98.67 | 97.70 | 99.65 | 95.74 | 98.67 | 97.70 | 96.72 | 98.67 | 97.50 |

Table 3: Comparison of our proposed `SFDA-PFT` method against state-of-the-art SFDA methods on the `BAH` dataset (10 target subjects, 214 source subjects). Bold numbers indicate the best `F1`.

| Setting | Methods | Sub-1 | Sub-2 | Sub-3 | Sub-4 | Sub-5 | Sub-6 | Sub-7 | Sub-8 | Sub-9 | Sub-10 | Average |
|---|---|---|---|---|---|---|---|---|---|---|---|---|
| Source-only | Source model | 11.20 | 17.84 | 12.60 | 18.50 | 14.10 | 16.92 | 10.30 | 13.40 | 16.00 | 15.31 | 14.62 |
| SFDA (model-based) | SHOT (Liang et al., 2020) | 40.53 | 47.91 | 42.14 | 46.20 | 39.81 | 48.52 | 41.02 | 45.70 | 44.23 | 45.13 | 44.10 |
| | NRC (Yang et al., 2021) | 48.72 | 42.30 | 46.00 | 44.10 | 41.81 | 47.58 | 43.71 | 44.65 | 47.93 | 44.12 | 45.00 |
| | TPDS (Tang et al., 2024) | 41.22 | 46.30 | 44.01 | 42.54 | 47.82 | 40.95 | 45.53 | 43.29 | 47.18 | 42.23 | 44.20 |
| | DSFDA (Sharafi et al., 2025) | 49.10 | 44.70 | 47.51 | 42.92 | 50.23 | 45.30 | 46.70 | 47.02 | 41.82 | 49.84 | 46.10 |
| SFDA (data-based) | SFIT (Hou & Zheng, 2021b) | 56.83 | 50.91 | 54.72 | 52.10 | 57.54 | 49.82 | 55.91 | 51.23 | 58.12 | 51.40 | 52.90 |
| | SFDA-IT (Hou & Zheng, 2021a) | 48.50 | 55.71 | 50.81 | 53.95 | 47.21 | 54.12 | 49.03 | 52.64 | 50.32 | **56.00** | 51.80 |
| | **SFDA-PFT (ours)** | **61.52** | **55.10** | **60.42** | **53.81** | **59.73** | **56.05** | **61.91** | **54.25** | **62.84** | 54.70 | **57.40** |
| Oracle | Supervised fine-tuning | 96.20 | 92.81 | 95.70 | 94.25 | 96.53 | 93.91 | 95.14 | 94.72 | 92.53 | 97.01 | 94.88 |

**Implementation Details:** Our `SFDA-PFT` model was implemented using PyTorch and conducts all experiments on a single NVIDIA A100-SXM4-40GB GPU. The source classifier is built on a `ResNet-18` backbone, followed by a classifier trained for binary expression recognition. Results obtained with other backbone architectures are provided in the Appendix. `ResNet-18` was selected as the feature extractor due to its widespread adoption in prior FER and domain adaptation works. During target adaptation, only the subject-adaptive layers of the translator are updated. The source backbone and classifier remain fixed. We train the model using the Adam optimizer with a learning rate of $1 \times 10^{-3}$ and a batch size of 64. A learning rate scheduler (ReduceLROnPlateau) was used, which monitors the validation loss and reduces the learning rate by a factor of 0.5 if no improvement is observed for 3 consecutive epochs. We set $\lambda_{\text{expr}} = 1.0$ and $\lambda_{\text{style}} = 0.3$, giving expression preservation higher priority while allowing the style loss to act as a regularizer for identity alignment.

## 4.2 COMPARISON WITH STATE-OF-THE-ART METHODS

For the lab-controlled datasets `BioVid` and `StressID`, `SFDA-PFT` achieves the highest `F1` among all methods (Table 1 and Table 2). On `BioVid`, which is relatively balanced across classes, `SFDA-PFT` obtains an average `F1` of 78.31, outperforming DSFDA by almost 10 points. More-

Table 4: Comparison of our proposed `SFDA-PFT` method against state-of-the-art SFDA methods on the `Aff-Wild2` dataset (10 target subjects, 282 source subjects). Bold numbers indicate the best `F1`.

| Setting | Methods | Sub-1 | Sub-2 | Sub-3 | Sub-4 | Sub-5 | Sub-6 | Sub-7 | Sub-8 | Sub-9 | Sub-10 | Average |
|---|---|---|---|---|---|---|---|---|---|---|---|---|
| Source-only | Source model (no adaptation) | 18.70 | 19.60 | 20.50 | 20.00 | 21.00 | 20.50 | 21.40 | 22.30 | 20.00 | 21.00 | 20.50 |
| SFDA (model-based) | SHOT (Liang et al., 2020) | 33.77 | 34.67 | 35.57 | 35.07 | 36.07 | 35.57 | 36.47 | 37.37 | 35.07 | 36.07 | 35.57 |
| | NRC (Yang et al., 2021) | 34.24 | 35.14 | 36.04 | 35.54 | 36.54 | 36.04 | 36.94 | 37.84 | 35.54 | 36.54 | 36.04 |
| | TPDS (Tang et al., 2024) | 36.69 | 37.59 | 38.49 | 37.99 | 38.99 | 38.49 | 39.39 | 40.29 | 37.99 | 38.99 | 38.49 |
| | DSFDA (Sharafi et al., 2025) | 37.26 | 38.16 | 39.06 | 38.56 | 39.56 | 39.06 | 39.96 | 40.86 | 38.56 | 39.56 | 39.06 |
| SFDA (data-based) | SFIT (Hou & Zheng, 2021b) | 48.43 | 49.33 | 50.23 | 49.73 | 50.73 | 50.23 | 51.13 | 52.03 | 49.73 | 50.73 | 50.23 |
| | SFDA-IT (Hou & Zheng, 2021a) | 49.30 | 50.20 | 51.10 | 50.60 | 51.60 | 51.10 | 52.00 | 52.90 | 50.60 | 51.60 | 51.10 |
| | **SFDA-PFT (ours)** | **52.66** | **53.56** | **54.46** | **53.96** | **54.96** | **54.46** | **55.36** | **56.26** | **53.96** | **54.96** | **54.46** |
| Oracle | Supervised fine-tuning | 91.93 | 92.83 | 93.73 | 93.23 | 94.23 | 93.73 | 94.63 | 95.53 | 93.23 | 94.23 | 93.73 |

over, to assess robustness across runs, we repeated `SFDA-PFT` training on `BioVid` with three independent seeds and obtained a stable average `F1` of 78.43 ± 0.25%, confirming low variance and consistent performance across random initializations. The main failure case is Sub-10, where `SFDA-PFT` drops to 58.08. A closer look shows that this subject is from an older age group, where pain-related facial reactions tend to be weaker and more varied. Because of this, the model struggles with recall, even though precision remains high. This indicates that age differences represent a challenge for personalization, consistent with prior FER studies reporting systematically lower recognition accuracy for older adults (Guo et al., 2013; SÖNMEZ, 2019; Kim et al., 2021a), and points to the value of age-aware or group-based adaptation strategies. On `StressID`, which is strongly imbalanced, `SFDA-PFT` reaches 69.92%, over 7 points higher than the best competing method, showing that it can handle skewed class distributions while still capturing subject-specific patterns. On the in-the-wild datasets `BAH` and `Aff-Wild2` (Table 3 and Table 4), class imbalance, noisy annotations, and uncontrolled acquisition conditions make `F1` a more reliable evaluation metric than accuracy. Here, `SFDA-PFT` again delivers the strongest performance, with 57.40% on `BAH` and 54.46% on `Aff-Wild2`, outperforming all alternatives. A key factor behind this improvement is that `SFDA-PFT` operates directly in the feature space, leveraging the robust representations already extracted by the backbone. In contrast, image-translation-based methods attempt to map target samples into a synthetic source domain, often introducing artifacts, blurring, or distortions that suppress subtle but critical expression cues such as micro-expressions or localized muscle activations. These imperfections propagate downstream and degrade classifier performance. By avoiding pixel-level synthesis, `SFDA-PFT` preserves discriminative structures in the feature space and provides more stable adaptation under the severe class imbalance and noise characteristic of real-world settings. Full accuracy results are reported in the Appendix, but we emphasize that `F1` is a more informative criterion in these imbalanced scenarios.

Table 5 compares `SFDA-PFT` with representative SFDA baselines in terms of accuracy and adaptation efficiency. We select baselines that, like our setting, update only a subset of parameters instead of fully fine-tuning the network. Here, *iterations* are the number of gradient update steps during test-time adaptation, and *time* is the average wall-clock adaptation time per batch. Despite requiring many more

Table 5: Comparison of SFDA models on `BioVid` in terms of accuracy, number of iterations, and convergence time per batch (B=64).

| Method | ACC (%) | Iters | Time (s) |
|---|---|---|---|
| SFDA-DE (Liang et al., 2020) | 62.88 | 1400 | 65.5 |
| TPDS (Tang et al., 2024) | 65.57 | 900 | 60.0 |
| SHOT (Liang et al., 2020) | 50.35 | 1155 | 54.0 |
| NRC (Yang et al., 2021) | 60.31 | 705 | 75.0 |
| **SFDA-PFT (ours)** | **82.46** | **135** | **0.95** |

iterations and longer runtimes per batch, these baselines achieve lower accuracy than `SFDA-PFT`. In contrast, `SFDA-PFT` achieves the best accuracy with far fewer updates and sub-second adaptation per batch, showing that feature-space adaptation is markedly more efficient and scalable.

## 5 ABLATION STUDIES

**Impact of Source Subject Pairing Strategies.** To study the effect of source subject pairing during translator pretraining, we evaluate three strategies: random, cosine-based, and landmark-based. In the cosine-based strategy, well-classified source samples are paired based on feature similarity in the

Figure 3: Source subject pairing on the `BioVid` dataset. (a) Examples of random, cosine-based, and landmark-based pairs. (b) Average `ACC`, with landmark-based pairing performing best.

Table 6: Average `ACC` (%) on `BioVid` using `SFDA-PFT`, ablating expression and style losses.

| Setting | $\lambda_e$ | $\lambda_s$ | Acc. (%) |
|---|---|---|---|
| No Losses | ✗ | ✗ | 68.62 |
| Style Loss | ✗ | ✓ | 70.10 |
| Expression Loss | ✓ | ✗ | 71.60 |
| Full Loss | ✓ | ✓ | **82.46** |

Table 7: Similarity of expression and identity branches on `BioVid`. Fixed expr. means the same expression, different subjects, while Fixed subj. means same subject, different expressions.

| Branch | Fixed expr. | Fixed subj. |
|---|---|---|
| Expression | 0.75 | 0.40 |
| Identity | 0.53 | 0.85 |

embedding space via cosine distance. The landmark-based strategy leverages facial geometry and pose: landmarks are aligned using Procrustes analysis, and head-pose vectors from OpenFace (Amos et al., 2016) provide orientation cues. As shown in Figure 3, cosine- and landmark-based pairing outperform random selection, with landmark-based achieving the highest average accuracy. For the elderly (60+) subject, an age-aware variant selecting younger expression-matched references improves the `F1` score by about +7%. Detailed per-subject results are provided in the Appendix.

**Expression and Identity Specialization in Embeddings.** We evaluate the specialization of identity and expression by computing cosine similarities for pairs with (i) the same emotion but different subjects, and (ii) the same subject but different emotions. As shown in Table 7, the expression branch maintains higher similarity for the same emotion (0.75) compared to different emotions (0.40), while the identity branch shows stronger similarity for the same subject (0.85) than for different subjects with the same emotion (0.53). These results show that the expression and identity branches capture primarily emotion and identity features, respectively, with some overlap between them.

**Target Sample Distribution Across Source Subjects.** To quantify the distribution of target samples and ensure the model doesn't overfit to a single source identity, we use the Nearest Source Prototype Histogram. This visualization shows the cosine similarity between target embeddings and source prototypes, assigning each target sample to the closest source subject. As shown in Figure 4, the histogram reveals a diverse distribution of target samples across multiple source subjects, rather than concentrating on one. This confirms that our model avoids overfitting, promoting better generalization while preserving the variation in identities and expressions across the source domain.

**Qualitative Analysis via t-SNE Visualization.** Figure 5 displays t-SNE plots for two complementary views: an *image-based* translation where target samples are first translated in pixel space and then embedded by the frozen backbone, and a *feature-based* translation of `SFDA-PFT` that operates directly on latent features. In the image-based plots, translated target clusters remain noticeably shifted away from the source clusters, revealing a larger residual domain gap. In contrast, the `SFDA-PFT` plots show target points tightly overlapping the source manifolds, indicating that latent-space translation induces a smaller domain shift while preserving the expression structure.

**Impact of Feature Vector Size on Performance** We conducted an ablation study to investigate the impact of feature dimensionality on the performance of feature translation across four FER datasets: `BioVid`, `StressID`, `BAH`, and `Aff-Wild2`. For each dataset, we varied the dimensionality of the translated feature vector from 64 to 512 and observed consistent improvements in accuracy with increasing dimensionality. Notably, the performance gains saturated around 256 or

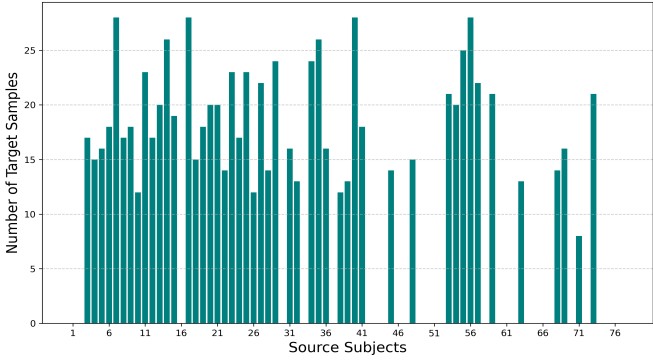

Figure 4: Distribution of target samples for sub-1 in `BioVid` dataset across source subjects

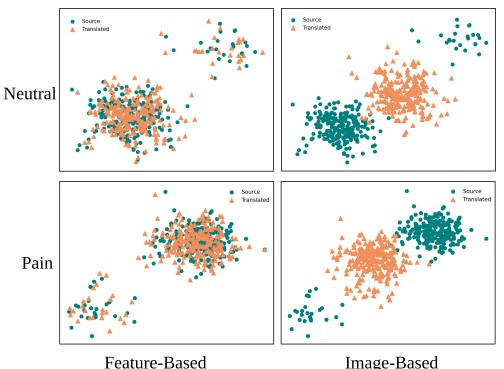

Figure 5: T-SNE of source vs. translated features for Sub-1 in `BioVid` comparing feature-based (left) and image-based (right) translation.

Figure 6: `SFDA-PFT` accuracy across feature dimensions (64–512) on four datasets, showing performance gains with higher dimensions.

512 dimensions, suggesting that higher-dimensional features provide richer identity and expression information. However, the marginal gains beyond 256 dimensions diminish, indicating a trade-off between representational power and computational efficiency. These trends are illustrated in Figure 6, highlighting the importance of selecting an appropriate feature size for effective and efficient.

## 6 CONCLUSION

This paper introduces `SFDA-PFT`, an efficient SFDA method tailored for personalization FER using only image data with neutral expressions from target subjects. Unlike traditional image-based approaches that depend on expressive target data and computationally expensive generative models, `SFDA-PFT` operates entirely in the feature space. It translates features from one subject to another in the source domain by aligning subject-specific features while preserving the expression of the original subject. This allows the model to maintain the expression of the input while adapting to the source subject, and to provide cost-effective personalization without requiring target expression data. The `SFDA-PFT` adaptation process involves adapting only a few layers of the translator module on the target subject's neutral data. `SFDA-PFT` is computationally efficient, stable during training, and well-suited for deployment in privacy-sensitive real-world scenarios such as healthcare or mobile applications. Experiments on four video FER datasets show that `SFDA-PFT` can achieve a higher level of performance with lower complexity, generalizing well across both controlled and in-the-wild conditions.

ACKNOWLEDGMENTS

This work was supported in part by the Fonds de recherche du Québec – Santé (FRQS), the Natural Sciences and Engineering Research Council of Canada (NSERC), Canada Foundation for Innovation (CFI), and the Digital Research Alliance of Canada.

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

# Appendix

This supplementary material provides additional insights and evidence supporting the main paper. It includes detailed descriptions of baseline methods, algorithmic procedures, extended experiments on additional datasets, ablation studies analyzing key components, and a summary of hyperparameter configurations used in our evaluations.

**Algorithm Details**

- Source Pre-training
- Target Adaptation

**Baseline Methods**

**Extended Experimental Results**

- Datasets
- Quantitative Comparison with SFDA Baselines
- Qualitative Examples of SFDA-IT Translations

**Additional Ablation Studies**

- Distance to Closest Source Prototypes
- Qualitative Analysis via t-SNE visualization
- Impact of Different Backbones
- Source Layer Selection for Style Transfer
- Effect of Expression Loss Type
- Sensitivity Analysis of Loss Weights.

**Hyperparameter Details**

---

**Algorithm 1** Source Pre-training

---

1: **procedure** PRETRAINSOURCE($\mathcal{D}_S, \mathbf{F}, \mathbf{C}, \mathbf{T}$)
2:     Initialize $\mathbf{F}, \mathbf{C}, \mathbf{T}$
3:     **for** each epoch **do**
4:         **for all** $(\mathbf{x}_s, y_s) \in \mathcal{D}_S$ **do**
5:             $\mathbf{f}_s \leftarrow \mathbf{F}(\mathbf{x}_s)$
6:             $y_{\text{pred}} \leftarrow \mathbf{C}(\mathbf{f}_s)$
7:             $\mathcal{L}_{\text{CE}} \leftarrow \text{CrossEntropy}(y_{\text{pred}}, y_s)$
8:             Update $\mathbf{F}, \mathbf{C}$ using $\mathcal{L}_{\text{CE}}$
9:         **end for**
10:     **end for**
11:     Freeze $\mathbf{F}$
12:     **for** each epoch **do**
13:         **for all** paired $(\mathbf{x}_1, y_1), (\mathbf{x}_2, \cdot) \in \mathcal{D}_S$ **do**
14:             $\mathbf{f}_1 \leftarrow \mathbf{F}(\mathbf{x}_1)$
15:             $\mathbf{f}_2 \leftarrow \mathbf{F}(\mathbf{x}_2)$
16:             $\hat{\mathbf{f}}_1 \leftarrow \mathbf{T}(\mathbf{f}_1)$
17:             $\mathcal{L}_{\text{expr}} \leftarrow D_{\text{KL}}(\mathbf{C}(\mathbf{f}_1) \,\|\, \mathbf{C}(\hat{\mathbf{f}}_1))$
18:             $\mathcal{L}_{\text{style}} \leftarrow 0$
19:             **for all** $l \in \mathcal{L}$ **do**
20:                 $\mu_1, \sigma_1 \leftarrow \text{MeanStd}(\hat{\mathbf{f}}_1^l)$
21:                 $\mu_2, \sigma_2 \leftarrow \text{MeanStd}(\mathbf{f}_2^l)$
22:                 $\mathcal{L}_{\text{style}} \leftarrow \mathcal{L}_{\text{style}} + \|\mu_1 - \mu_2\|^2 + \|\sigma_1 - \sigma_2\|^2$
23:             **end for**
24:             $\mathcal{L}_{\text{CE}} \leftarrow \text{CrossEntropy}(\mathbf{C}(\hat{\mathbf{f}}_1), y_1)$
25:             $\mathcal{L}_{\text{total}} \leftarrow \mathcal{L}_{\text{CE}} + \lambda_{\text{expr}} \cdot \mathcal{L}_{\text{expr}} + \lambda_{\text{style}} \cdot \mathcal{L}_{\text{style}}$
26:             Update $\mathbf{T}$ using $\mathcal{L}_{\text{total}}$
27:         **end for**
28:     **end for**
29: **end procedure**

---

## A  ALGORITHM DETAILS

This section outlines the core procedures of our proposed Personalized Feature Translation (`SFDA-PFT`) framework for SFDA. The method comprises two stages: (1) source pre-training, and (2) target-domain. The full pseudocode is provided in Algorithms 1 and 2, and we define the main notations below.

**Architecture:** Let $\mathcal{D}_S = \{(\mathbf{x}_s, y_s)\}$ be a labeled source dataset, where $\mathbf{x}_s$ is a source subject and $y_s \in \mathcal{Y}$ its corresponding expression label. Let $\mathcal{D}_T = \{\mathbf{x}_t\}$ denote the unlabeled dataset for a target subject. We denote by $\mathbf{F}$ the source feature extractor and by $\mathbf{C}$ the classifier head. The translator network is defined as the composition of $\mathbf{F}$ followed by a set of lightweight, subject-adaptive layers $\mathbf{T}$. Thus, the translator $\mathbf{T}_{\text{full}} = \mathbf{T} \circ \mathbf{F}$ takes an image as input and outputs a translated feature representation. The source classifier $(\mathbf{F}, \mathbf{C})$ is trained on $\mathcal{D}_S$ and remains frozen during adaptation. The translator is first pretrained on $\mathcal{D}_S$ to learn identity transformation while preserving expression, and then adapted to each target subject individually using only a few samples.

## B  BASELINE METHOD DESCRIPTIONS

We compare our proposed method against seven representative SFDA baselines. These methods span both feature-space and image-space adaptation strategies, enabling a comprehensive evaluation of our approach. For fairness, all baselines are implemented using a fixed `ResNet-18` backbone and evaluated under a consistent experimental protocol.

- **SHOT** (Liang et al., 2020) freezes the source feature extractor and adapts only the classifier using pseudo-labeling and information maximization, encouraging discriminative clustering in the target domain without accessing source data.

---

**Algorithm 2** Target Adaptation

---

1: **procedure** ADAPTTOTARGET($\mathcal{D}_T, \mathbf{F}, \mathbf{C}, \mathbf{T}$)
2:     Freeze $\mathbf{F}, \mathbf{C}$
3:     **for** each epoch **do**
4:         **for all** $\mathbf{x}_t \in \mathcal{D}_T$ **do**
5:             $\mathbf{f}_t \leftarrow \mathbf{F}(\mathbf{x}_t)$
6:             $\hat{\mathbf{f}}_t \leftarrow \mathbf{T}(\mathbf{f}_t)$
7:             $\mathcal{L}_{\text{expr}} \leftarrow D_{\text{KL}}(\mathbf{C}(\mathbf{f}_t) \,\|\, \mathbf{C}(\hat{\mathbf{f}}_t))$
8:             Update $\mathbf{T}$ using $\mathcal{L}_{\text{expr}}$
9:         **end for**
10:    **end for**
11: **end procedure**

---

- **TPDS** (Tang et al., 2024) introduces a progressive adaptation framework that bridges source and target domains via a series of proxy distributions, aligning predictions using category consistency and mutual information objectives.

- **NRC** (Yang et al., 2021) exploits the intrinsic neighborhood structure of the target data by enforcing label consistency among reciprocal neighbors, using a memory bank for efficient retrieval.

- **DSFDA** (Sharafi et al., 2025) adapts FER models using only neutral target videos by disentangling identity and expression features, generating synthetic expressive data, and jointly training in a one-stage framework.

- **SFIT** (Hou & Zheng, 2021b) visualizes the knowledge gap between source and target models by translating target images into source-style images using only the two model checkpoints. It employs a generator guided by knowledge distillation and a relationship-preserving loss, enabling adaptation and fine-tuning without source data.

- **SFDA-IT** (Hou & Zheng, 2021a) formulates domain adaptation as an image translation problem where a generator maps target images into source-style images without paired supervision. The translated images are then classified by the fixed source model, improving performance through batch-wise style alignment and entropy regularization.

## C   EXTENDED EXPERIMENTAL RESULTS

### C.1  DATASETS

- `BioVid`: Heat and Pain (Part A): This dataset (Walter et al., 2013) consists of video recordings of 87 subjects experiencing thermal pain stimuli in a controlled laboratory setting. Each subject is assigned to one of five pain categories: "no pain" and four increasing pain levels (PA1–PA4), with PA4 representing the highest intensity. Consistent with prior work, which reports minimal facial activity at lower intensities, we focus on a binary classification between "no pain" and PA4. For each subject, 20 videos per class are used, each lasting 5.5 seconds. Following recommendations in (Werner et al., 2017), the first 2 seconds of each PA4 video are discarded to eliminate frames where facial expressions are typically absent, retaining only the segments that capture stronger pain-related facial activity.

- `StressID`: This dataset (Chaptoukaev et al., 2023) focuses on assessing stress through facial expressions. It comprises facial video recordings from 54 individuals, totaling around 918 minutes of annotated visual content. In our work, we use only the visual modality. Each frame is labeled as either "neutral" or "stressed," based on participants' self-reported stress scores. Specifically, frames corresponding to scores below 5 are labeled as neutral (label 0), while those with scores of 5 or higher are considered stressed (label 1).

- `BAH`: The `BAH` dataset (González-González et al., 2025), which is designed for recognizing ambivalence and hesitancy (A/H) expressions in real-world video recordings.comprises facial recordings from 224 participants across Canada, designed to reflect a diverse demographic distribution in terms of sex, ethnicity, and province. Each participant contributes up to seven videos, with a total

Table 8: Comparison of the proposed `SFDA-PFT` with state-of-the-art SFDA methods on the `BioVid` dataset (10 target subjects, 77 source subjects). All models use `ResNet-18`. Bold numbers indicate the best `ACC`.

| Setting | Methods | Sub-1 | Sub-2 | Sub-3 | Sub-4 | Sub-5 | Sub-6 | Sub-7 | Sub-8 | Sub-9 | Sub-10 | Average |
|---|---|---|---|---|---|---|---|---|---|---|---|---|
| Source-only | Source model (no adaptation) | 66.11 | 55.55 | 86.36 | 85.11 | 87.11 | 59.00 | 75.66 | 70.44 | 52.66 | 48.22 | 68.62 |
| SFDA (model-based) | SHOT (Liang et al., 2020) | 55.78 | 47.76 | 41.05 | 52.44 | 54.67 | 48.89 | 54.46 | 49.22 | 54.86 | 44.44 | 50.35 |
| | NRC (Yang et al., 2021) | 59.33 | 39.38 | 84.00 | 72.89 | 79.67 | 42.67 | 63.92 | 53.95 | 54.89 | 42.47 | 60.31 |
| | TPDS (Tang et al., 2024) | 65.56 | 55.98 | 79.22 | 68.22 | 91.67 | 59.11 | 61.28 | 69.33 | 57.11 | 48.22 | 65.57 |
| | DSFDA (Sharafi et al., 2025) | 77.00 | 75.11 | 90.89 | 85.67 | 88.11 | 67.48 | 89.22 | **87.56** | 69.22 | 72.11 | 80.24 |
| SFDA (data-based) | SFIT (Hou & Zheng, 2021b) | 80.92 | 68.79 | 82.87 | 84.70 | 91.62 | 57.32 | 60.59 | 74.56 | 60.71 | **79.94** | 74.20 |
| | SFDA-IT (Hou & Zheng, 2021a) | 75.33 | 67.27 | 89.00 | 84.55 | 90.80 | 62.31 | 81.77 | 75.89 | 57.88 | 70.56 | 75.54 |
| | **SFDA-PFT (ours)** | **84.93** | **75.56** | **95.05** | **85.86** | **97.59** | **73.78** | **88.73** | 83.49 | **78.48** | 61.16 | **82.46** |
| Oracle | Fine-Tuning | 97.11 | 91.43 | 96.76 | 97.89 | 96.11 | 92.30 | 90.01 | 95.09 | 98.22 | 97.00 | 95.19 |

of 1,118 videos ( 86.2 hours). Among these, 638 videos contain at least one A/H segment, resulting in a total of 1,274 annotated A/H segments. The dataset includes 143,103 frames labeled with A/H, out of 714,005 total frames. In our setup, frames with A/H annotations are assigned a label of 1 (indicating the presence of A or H), while all other frames are considered neutral and assigned a label of 0.

- `Aff-Wild2`: The `Aff-Wild2` dataset (Kollias & Zafeiriou, 2019) is a large-scale in-the-wild dataset for affect recognition, consisting of 318 videos with available annotations. In our study, we use a subset of 292 videos that each represent a single subject, which is essential for our subject-based setting, where each individual is treated as a separate domain. We focus exclusively on basic expression categories for discrete expression classification. Specifically, we use the following seven classes: neutral (0), anger (1), disgust (2), fear (3), happiness (4), sadness (5), and surprise (6). We consider only the visual modality in our experiments.

## C.2 QUANTITATIVE COMPARISON WITH SFDA BASELINES

Across the lab-controlled datasets `BioVid` and `StressID`, our proposed `SFDA-PFT` achieves the best performance compared to all baselines (Table 8 and Table 9). On `BioVid`, `SFDA-PFT` improves the average accuracy to 82.46%, surpassing the best baseline (SFDA-TT) by more than 2 percentage points. Similarly, on `StressID`, `SFDA-PFT` reaches 71.49%, again outperforming all competing methods. These gains highlight the advantage of `SFDA-PFT` in settings where acquisition conditions are stable, allowing feature-level adaptation to capture subtle subject-specific differences with reduced variance across individuals. While overall performance is strong, two notable failure cases appear on `BioVid` for Sub-8 and Sub-10, where `SFDA-PFT` achieves only 83.49% and 61.16%, respectively. Both subjects belong to the older age group, where pain-related facial responses are less pronounced and more variable, reducing the discriminability of features. This suggests that subject age can act as a confounding factor in personalized adaptation and points to the potential benefit of future age-aware or stratified domain adaptation strategies.

On the more in-the-wild datasets `BAH` and `Aff-Wild2`, `SFDA-PFT` remains highly competitive (Table 10 and Table 11). On `BAH`, `SFDA-PFT` clearly outperforms all alternatives, achieving 62.09% average accuracy, over 4 points higher than the best image-translation baseline. On `Aff-Wild2`, which involves 7 classes and severe real-world noise, `SFDA-PFT` performs on par with the strongest baseline, trailing by less than 1 percentage point. The remaining gap arises from multi-class confusion and extreme conditions such as pose variation, motion blur, and class imbalance. Notably, `SFDA-PFT` surpasses image-translation-based methods because it adapts directly in the feature space rather than the image space: image translation often introduces artifacts or loses discriminative details (e.g., subtle muscle activations or micro-expressions), which weakens downstream classification. By preserving the discriminative structure already extracted by the backbone, `SFDA-PFT` avoids error accumulation from imperfect translations and provides more stable, reliable adaptation across subjects.

Table 9: Comparison of the proposed `SFDA-PFT` with state-of-the-art SFDA methods on the `StressID` dataset (10 target subjects, 44 source subjects). All models use `ResNet-18`. Bold numbers indicate the best `ACC`.

| Setting | Methods | Sub-1 | Sub-2 | Sub-3 | Sub-4 | Sub-5 | Sub-6 | Sub-7 | Sub-8 | Sub-9 | Sub-10 | Average |
|---|---|---|---|---|---|---|---|---|---|---|---|---|
| Source-only | Source model (no adaptation) | 38.96 | 41.21 | 65.53 | 42.04 | 55.16 | 65.51 | 69.43 | 60.78 | 53.62 | 55.63 | 54.79 |
| SFDA (model-based) | SHOT (Liang et al., 2020) | 68.33 | 51.95 | 45.83 | 39.26 | 53.67 | 61.38 | 59.76 | 45.25 | 51.42 | 52.05 | 52.88 |
| | NRC (Yang et al., 2021) | 69.03 | 52.25 | 31.83 | 35.29 | 59.67 | 42.50 | 59.28 | 41.25 | 65.42 | 54.20 | 51.07 |
| | TPDS (Tang et al., 2024) | 65.56 | 54.98 | 64.22 | 58.22 | 54.67 | 63.11 | 69.28 | 59.33 | 50.11 | 51.98 | 59.17 |
| | DSFDA Sharafi et al. (2025) | 73.47 | 69.39 | **87.12** | 69.74 | 79.87 | 87.39 | 82.80 | 83.89 | **75.03** | 77.39 | 78.61 |
| SFDA (data-based) | SFIT (Hou & Zheng, 2021b) | 70.41 | 68.85 | 69.67 | 71.92 | 67.48 | 77.43 | 70.76 | 75.21 | 65.98 | 61.19 | 69.89 |
| | SFDA-IT (Hou & Zheng, 2021a) | 73.47 | 69.50 | 69.90 | 73.02 | 66.54 | 78.62 | 71.30 | 76.67 | 65.42 | 67.32 | 71.18 |
| | **SFDA-PFT (ours)** | **78.33** | **74.87** | 78.17 | **73.32** | **79.96** | **89.00** | **84.76** | **84.14** | 74.42 | **77.95** | **79.49** |
| Oracle | Fine-Tuning | 98.89 | 100 | 99.53 | 98.15 | 99.22 | 97.57 | 96.02 | 99.38 | 99.56 | 100 | 98.83 |

Table 10: Comparison between the proposed `SFDA-PFT` and several state-of-the-art methods on the `BAH` dataset (10 target subjects, 214 source subjects). All models use `ResNet-18`. Bold numbers indicate the best `ACC`.

| Setting | Methods | Sub-1 | Sub-2 | Sub-3 | Sub-4 | Sub-5 | Sub-6 | Sub-7 | Sub-8 | Sub-9 | Sub-10 | Average |
|---|---|---|---|---|---|---|---|---|---|---|---|---|
| Source-only | Source model | 49.71 | 50.00 | 54.67 | 47.71 | 48.43 | 51.51 | 48.83 | 50.30 | 50.45 | 48.65 | 50.03 |
| SFDA (model-based) | SHOT (Liang et al., 2020) | 49.73 | 54.17 | **60.49** | 49.55 | 45.83 | 51.22 | 46.62 | 52.76 | 49.09 | 47.92 | 50.74 |
| | NRC (Yang et al., 2021) | 49.46 | 54.05 | 55.02 | 49.11 | 46.52 | 49.91 | 44.83 | 52.26 | 48.63 | 47.24 | 49.70 |
| | TPDS (Tang et al., 2024) | 50.42 | 52.38 | 55.91 | 48.75 | 47.67 | 51.66 | 44.83 | 53.20 | 51.75 | 55.13 | 51.17 |
| | DSFDA (Sharafi et al., 2025) | 61.24 | 56.02 | 60.31 | 58.77 | 54.19 | 59.88 | **57.40** | 53.16 | 62.15 | 60.49 | 58.36 |
| SFDA (data-based) | SFIT (Hou & Zheng, 2021b) | 60.15 | 56.84 | 60.04 | 54.91 | 56.15 | 55.73 | 56.02 | 56.49 | 56.22 | 58.39 | 57.09 |
| | SFDA-IT (Hou & Zheng, 2021a) | 60.00 | 60.12 | 56.48 | 55.73 | 56.15 | 57.42 | 56.80 | 55.96 | 56.89 | 57.05 | 57.26 |
| | **SFDA-PFT (ours)** | **69.46** | **64.17** | 60.49 | **62.11** | **59.83** | **61.91** | 54.62 | **57.76** | **62.63** | **67.92** | **62.09** |
| Oracle | Fine-tune | 93.35 | 96.61 | 99.22 | 95.58 | 99.17 | 97.89 | 92.48 | 96.14 | 93.07 | 93.38 | 95.69 |

Table 11: Comparison between the proposed `SFDA-PFT` and several state-of-the-art methods on the `Aff-Wild2` dataset (10 target subjects, 282 source subjects). All models use `ResNet-18`. Bold numbers indicate the best `ACC`.

| Setting | Methods | Sub-1 | Sub-2 | Sub-3 | Sub-4 | Sub-5 | Sub-6 | Sub-7 | Sub-8 | Sub-9 | Sub-10 | Average |
|---|---|---|---|---|---|---|---|---|---|---|---|---|
| Source-only | Source model | 24.50 | 23.97 | 21.94 | 30.17 | 32.41 | 40.63 | 16.92 | 37.67 | 18.77 | 19.98 | 26.70 |
| SFDA (model-based) | SHOT (Liang et al., 2020) | 45.00 | 41.67 | 40.42 | 43.91 | 39.88 | 41.34 | 42.18 | 39.00 | 49.16 | 50.84 | 42.34 |
| | NRC (Yang et al., 2021) | 43.77 | 42.31 | 40.98 | 44.26 | 41.83 | 40.61 | 42.12 | 43.29 | 49.12 | 50.71 | 42.90 |
| | TPDS (Tang et al., 2024) | 47.62 | 44.18 | 43.75 | 46.03 | 42.87 | 41.59 | 44.22 | 41.07 | 49.08 | 50.49 | 45.09 |
| | DSFDA (Sharafi et al., 2025) | 58.31 | 56.78 | 57.96 | 59.24 | 55.63 | 58.09 | 56.41 | 57.18 | 57.82 | 56.18 | 57.42 |
| SFDA (data-based) | SFIT (Hou & Zheng, 2021b) | 58.42 | 56.37 | 54.79 | 57.61 | 55.03 | 52.98 | 56.85 | 57.12 | 49.56 | 50.57 | 55.93 |
| | SFDA-IT (Hou & Zheng, 2021a) | 59.63 | 57.88 | 54.42 | 58.07 | 53.61 | 55.94 | 50.83 | 55.76 | 58.23 | **56.83** | 56.12 |
| | **SFDA-PFT (ours)** | **60.83** | **59.47** | **58.26** | **61.72** | **57.39** | **60.91** | 56.18 | **59.64** | **61.05** | 56.75 | **59.20** |
| Oracle | Fine-tune | 98.90 | 98.71 | 98.03 | 98.37 | 94.66 | 83.33 | 99.87 | 81.48 | 94.54 | 97.88 | 94.58 |

### C.3 QUALITATIVE EXAMPLES OF SFDA-IT TRANSLATIONS

In addition to quantitative results, we also provide qualitative examples of translated images generated by SFDA-IT (Hou & Zheng, 2021a). As an image-based adaptation method, SFDA-IT (Hou & Zheng, 2021a) maps target-domain samples into a source-style visual space before classification. Figure 7 illustrates representative examples from `BioVid`, showing target input frames and their translated counterparts. While SFDA-IT (Hou & Zheng, 2021a) effectively alters low-level style features, it may fail to preserve fine-grained facial expressions essential for accurate classification, particularly in subtle affective states.

## D ADDITIONAL ABLATION STUDIES

### D.1 ISTANCE TO CLOSEST SOURCE PROTOTYPES

To assess the effectiveness of our subject-aware translation module, we plot the L2 distances between translated target samples and the closest source subject prototype in the feature space. As

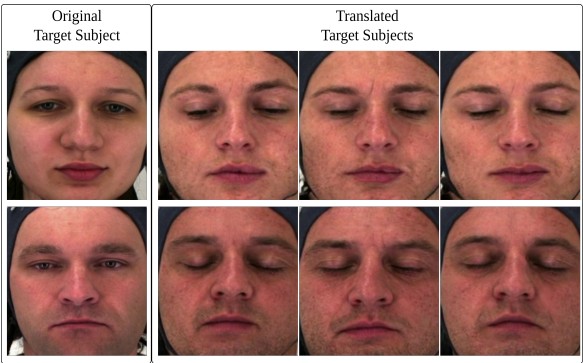

Figure 7: Translated images of two target subjects from the `BioVid` dataset using landmark pairs at test time with the SFDA-IT (Hou & Zheng, 2021a) method. *Left* column shows the original target image. *Right* columns display the corresponding translated images used for classification.

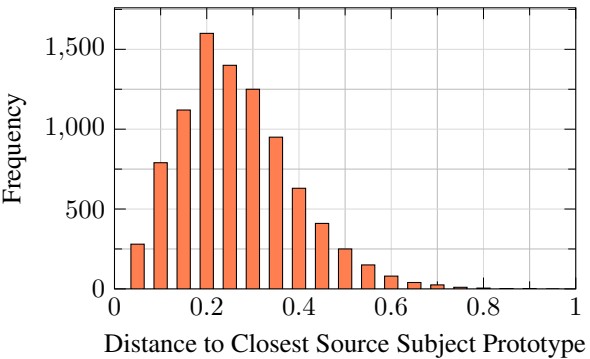

Figure 8: Histogram of distances between translated target frames and their closest source prototype.

Figure 9: t-SNE visualizations before and after `SFDA-PFT` adaptation for Sub-1 in `BioVid` dataset.

shown in Figure 8, the distribution is asymmetric, with a strong concentration of distances around 0.2 and a long tail toward higher values. This indicates that most translated target features are successfully aligned close to their corresponding source subject representations, validating the role of the translation mechanism in enhancing subject-level alignment. The sharp peak and reduced spread reflect improved intra-class compactness and inter-domain consistency, which are critical for minimizing domain shift in source-free adaptation settings.

### D.2 QUALITATIVE ANALYSIS VIA T-SNE VISUALIZATION.

Target embeddings are visualized using t-SNE for a representative subject (Sub-1) across four models, source-only, SFIT (Hou & Zheng, 2021b)(b), SFDA-IT (Hou & Zheng, 2021a), and our `SFDA-PFT` (Figure. 9). Initially, the source-only model (a) yields overlapping neutral/pain clusters. After adaptation, SFIT (b) adds mild structure but remains mixed; SFDA-IT (c) shows clearer yet diffuse boundaries; `SFDA-PFT` (d) forms compact, well-separated clusters, indicating better expression preservation and domain alignment.

### D.3 IMPACT OF DIFFERENT BACKBONES

To assess whether the gains of `SFDA-PFT` depend on a particular backbone, we repeat all experiments with both a transformer encoder (`ViT-B/32 `) and a convolutional encoder (`ResNet-50`) across `BioVid`, `StressID`, and `BAH`. In all settings, `SFDA-PFT` achieves the best average `F1` and yields the highest or near-highest subject-wise performance (Tables 12–17), outperforming SHOT, DSFDA, SFIT, and SFDA-IT. These results indicate that the proposed feature-space translation is

Table 12: Comparison of the proposed `SFDA-PFT` with state-of-the-art SFDA methods on the `BioVid` dataset (10 target subjects, 77 source subjects) with `ViT-B/32` . Bold numbers indicate the best `F1`.

| Methods | Sub-1 | Sub-2 | Sub-3 | Sub-4 | Sub-5 | Sub-6 | Sub-7 | Sub-8 | Sub-9 | Sub-10 | Average |
|---|---|---|---|---|---|---|---|---|---|---|---|
| Src_only | 68.89 | 52.00 | 85.00 | 74.56 | 75.00 | 56.69 | 65.89 | 56.00 | 64.55 | 59.55 | 65.81 |
| SHOT | 69.99 | 66.00 | 82.14 | 75.69 | 79.69 | 57.00 | 69.00 | 74.69 | 69.00 | 61.60 | 70.48 |
| DSFDA | 70.16 | 66.52 | 85.99 | 81.11 | 85.66 | 64.36 | 78.22 | 75.00 | 67.23 | 63.00 | 73.72 |
| SFIT | 70.16 | 69.00 | 86.55 | 80.69 | 81.45 | 63.60 | 80.00 | 74.66 | 65.36 | 61.00 | 73.24 |
| SFDA-IT | 69.99 | 71.00 | 88.00 | 80.90 | 86.97 | 66.00 | 79.65 | 64.56 | 66.47 | 60.98 | 73.45 |
| **SFDA-PFT (ours)** | **77.00** | **71.00** | **90.20** | **81.50** | **88.10** | **67.50** | **84.60** | **75.20** | **68.00** | **63.40** | **76.65** |

Table 13: Comparison of the proposed `SFDA-PFT` with state-of-the-art SFDA methods on the `StressID` dataset (10 target subjects, 44 source subjects) with `ViT-B/32` . Bold numbers indicate the best `F1`.

| Methods | Sub-1 | Sub-2 | Sub-3 | Sub-4 | Sub-5 | Sub-6 | Sub-7 | Sub-8 | Sub-9 | Sub-10 | Average |
|---|---|---|---|---|---|---|---|---|---|---|---|
| Src_only | 47.10 | 35.56 | 58.12 | 50.98 | 51.28 | 38.76 | 45.05 | 38.29 | 44.14 | 40.72 | 45.00 |
| SHOT | 44.38 | 41.85 | 52.08 | 47.99 | 50.53 | 36.14 | 43.75 | 47.35 | 43.75 | 39.09 | 44.69 |
| DSFDA | 62.28 | 59.07 | 76.36 | 72.00 | 76.05 | 57.16 | 69.43 | 66.61 | **61.52** | 55.95 | 65.66 |
| SFIT | 59.35 | 58.40 | 73.29 | 68.28 | 68.95 | 53.84 | 67.72 | 63.15 | 55.29 | 51.63 | 62.00 |
| SFDA-IT | 61.18 | 62.58 | 76.87 | 70.63 | **78.07** | 57.65 | 69.57 | 56.39 | 58.06 | 53.27 | 64.43 |
| **SFDA-PFT (ours)** | **68.23** | **62.91** | **79.93** | **72.22** | **78.07** | **59.81** | **74.96** | **66.64** | 60.26 | **56.18** | **67.92** |

Table 14: Comparison of the proposed `SFDA-PFT` with state-of-the-art SFDA methods on the `BAH` dataset (10 target subjects, 214 source subjects) with `ViT-B/32` . Bold numbers indicate the best `F1`.

| Methods | Sub-1 | Sub-2 | Sub-3 | Sub-4 | Sub-5 | Sub-6 | Sub-7 | Sub-8 | Sub-9 | Sub-10 | Average |
|---|---|---|---|---|---|---|---|---|---|---|---|
| Src_only | 17.09 | 12.90 | 21.09 | 18.50 | 18.61 | 14.07 | 16.35 | 13.90 | 16.02 | 14.78 | 16.33 |
| SHOT | 42.68 | 40.25 | 50.09 | 46.16 | 48.60 | 34.76 | 42.08 | 45.55 | 42.08 | 37.56 | 42.98 |
| DSFDA | 45.23 | 42.89 | 55.44 | 52.29 | 55.23 | 41.49 | 50.43 | 48.35 | 44.63 | 40.62 | 47.66 |
| SFIT | 47.89 | 47.10 | 59.08 | 55.08 | 55.60 | 43.41 | 54.61 | 50.96 | 44.62 | 41.64 | 50.00 |
| SFDA-IT | 48.22 | 49.39 | 60.62 | 55.73 | 59.92 | 45.47 | 54.87 | 44.48 | 45.79 | 42.01 | 50.65 |
| **SFDA-PFT (ours)** | **55.75** | **51.41** | **65.31** | **59.01** | **63.79** | **48.87** | **61.26** | **54.45** | **49.24** | **45.91** | **55.50** |

Table 15: Comparison of the proposed `SFDA-PFT` with state-of-the-art SFDA methods on the `BioVid` dataset (10 target subjects, 77 source subjects) with `ResNet-50`. Bold numbers indicate the best `F1`.

| Methods | Sub-1 | Sub-2 | Sub-3 | Sub-4 | Sub-5 | Sub-6 | Sub-7 | Sub-8 | Sub-9 | Sub-10 | Average |
|---|---|---|---|---|---|---|---|---|---|---|---|
| Src_only | 65.00 | 53.23 | 79.56 | 68.88 | 72.66 | 53.00 | 65.00 | 55.66 | 63.22 | 54.69 | 63.09 |
| SHOT | 70.00 | 65.89 | 87.50 | 78.22 | 79.11 | 61.23 | 73.56 | 67.77 | 64.55 | 56.00 | 70.38 |
| DSFDA | 70.89 | 68.90 | 87.00 | 80.00 | 79.76 | 62.55 | 75.47 | 70.00 | 64.76 | 55.50 | 71.48 |
| SFIT | 75.02 | 69.00 | 88.65 | 80.00 | 80.14 | 62.50 | 81.95 | 72.45 | 64.80 | 60.00 | 73.45 |
| SFDA-IT | 76.25 | 69.00 | 88.65 | 80.69 | 80.00 | 62.00 | 81.50 | 72.45 | 65.80 | 60.40 | 73.67 |
| **SFDA-PFT (ours)** | **80.00** | **71.50** | **90.20** | **81.50** | **88.10** | **67.50** | **84.60** | **75.20** | **68.00** | **63.40** | **77.00** |

Table 16: Comparison of the proposed `SFDA-PFT` with state-of-the-art SFDA methods on the `StressID` dataset (10 target subjects, 44 source subjects) with `ResNet-50`. Bold numbers indicate the best `F1`.

| Methods | Sub-1 | Sub-2 | Sub-3 | Sub-4 | Sub-5 | Sub-6 | Sub-7 | Sub-8 | Sub-9 | Sub-10 | Average |
|---|---|---|---|---|---|---|---|---|---|---|---|
| Src_only | 49.22 | 37.00 | 56.54 | 51.11 | 51.28 | 40.69 | 50.89 | 40.18 | 42.56 | 45.44 | 46.49 |
| SHOT | 50.69 | 42.89 | 58.00 | 52.00 | 51.99 | 46.14 | 51.26 | 45.99 | 44.05 | 47.04 | 49.01 |
| DSFDA | 63.00 | 57.00 | 75.00 | 71.97 | 75.99 | 52.78 | 57.77 | **69.00** | 53.66 | 46.44 | 62.26 |
| SFIT | 58.69 | 57.46 | 73.00 | 69.78 | 76.00 | 57.36 | 69.69 | 65.89 | 57.69 | 52.00 | 63.76 |
| SFDA-IT | 61.00 | 60.55 | 75.75 | 70.00 | 77.11 | 56.65 | 70.07 | 66.30 | 58.66 | 55.00 | 65.11 |
| **SFDA-PFT (ours)** | **67.22** | **63.91** | **77.06** | **72.22** | **79.00** | **60.89** | **76.66** | 67.60 | **59.99** | **58.00** | **68.26** |

robust to the choice of backbone and can be used as a plug-and-play SFDA module for both CNN- and ViT-based FER models.

Table 17: Comparison of the proposed `SFDA-PFT` with state-of-the-art SFDA methods on the `BAH` dataset (10 target subjects, 214 source subjects) with `ResNet-50`. Bold numbers indicate the best F1.

| Methods | Sub-1 | Sub-2 | Sub-3 | Sub-4 | Sub-5 | Sub-6 | Sub-7 | Sub-8 | Sub-9 | Sub-10 | Average |
|---|---|---|---|---|---|---|---|---|---|---|---|
| Src_only | 16.00 | 12.07 | 23.11 | 18.01 | 18.96 | 14.99 | 17.22 | 13.44 | 16.09 | 15.02 | 16.49 |
| SHOT | 42.98 | 40.69 | 51.78 | 48.00 | 48.55 | 40.00 | 45.87 | 44.50 | 43.88 | 39.99 | 44.62 |
| DSFDA | 45.44 | 44.00 | 58.97 | 54.01 | 54.00 | 43.33 | 51.12 | 48.99 | 45.00 | 40.19 | 48.51 |
| SFIT | 45.76 | 46.66 | 59.28 | 56.01 | 55.33 | 44.98 | 54.00 | 49.15 | 44.25 | 41.14 | 49.66 |
| SFDA-IT | 45.95 | 47.00 | 61.11 | 55.22 | 55.58 | 45.45 | 58.11 | 49.88 | 46.08 | 43.00 | 50.74 |
| **SFDA-PFT (ours)** | **53.98** | **53.00** | **65.33** | **57.77** | **64.77** | **52.14** | **61.00** | **54.68** | **48.76** | **45.00** | **55.64** |

Table 18: Impact of source layer selection for style transfer on classification accuracy (%) using our proposed `SFDA-PFT` method for `BioVid` dataset.

| Layer Configuration | Accuracy (%) |
|---|---|
| Layer 1 | 74.13 |
| Layers 1–2 | 76.37 |
| Layers 1–3 | **82.46** |
| Last Layers | 69.25 |

### D.4 SOURCE LAYER SELECTION FOR STYLE TRANSFER

To assess the impact of style extraction depth, we experiment with using mean and variance statistics from different layers of the source model to transfer identity-specific information. As shown in Table 18, utilizing only early layers (e.g., Layer 1) yields moderate performance, while progressively including Layers 2 and 3 leads to significant improvements. This suggests that intermediate layers better capture subject-specific style without entangling high-level semantic content. In contrast, using the last layers results in a drop in accuracy, likely due to the abstraction of expression-related features. Overall, these findings highlight the importance of selecting appropriate layers for effective style modeling in source-free FER.

### D.5 EFFECT OF EXPRESSION LOSS TYPE

To evaluate the impact of different expression loss formulations, we compare mean squared error (MSE), cross-entropy (CE), and Kullback–Leibler (KL) divergence in both image-based and feature-based settings. As shown in Table 19, the feature-based model consistently outperforms its image-based counterpart across all loss types, further validating the advantages of operating in the latent feature space. Among the expression loss variants, KL divergence achieves the highest accuracy in both models, suggesting its strength in aligning soft expression distributions more effectively than point-wise (MSE) or hard-target (CE) alternatives. Notably, the feature-based model with KL divergence reaches 80.54% accuracy, outperforming the best image-based counterpart by over 5%, while also benefiting from reduced training cost and model complexity.

### D.6 SENSITIVITY ANALYSIS OF LOSS WEIGHTS.

We analyze the impact of the expression and style losses during source training and their effect on average target classification performance on `BioVid` dataset. As shown in Figure 10, turning off either loss (i.e., setting $\lambda_{expr}=0$ or $\lambda_{style}=0$) leads to a substantial drop in accuracy compared to the joint setting, confirming that both components are important for effective translation. The degradation is more pronounced when the style loss is removed, highlighting the dominant role of identity alignment for subject-specific adaptation. At the same time, varying $\lambda_{expr}$ (with $\lambda_{style}=0.3$) or $\lambda_{style}$ (with $\lambda_{expr}=1.0$) over a broad range yields only modest changes in performance, indicating that `SFDA-PFT` is not overly sensitive to moderate perturbations of these hyperparameters around the chosen setting ($\lambda_{expr}=1.0$, $\lambda_{style}=0.3$).

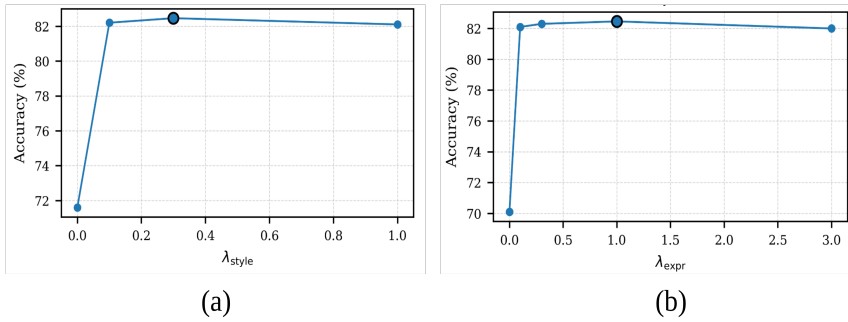

Figure 10: Sensitivity of our `SFDA-PFT` method to the expression and style loss weights on `BioVid`: (a) varying $\lambda_{\text{expr}}$ with $\lambda_{\text{style}}$=0.3; (b) varying $\lambda_{\text{style}}$ with $\lambda_{\text{expr}}$=1.0.

Table 19: Average target-domain classification accuracy (%) of SFDA-IT (Hou & Zheng, 2021a) and our proposed `SFDA-PFT` methods using different expression loss functions on the `BioVid` dataset.

| expression Loss Type | Image-based | Feature-based |
|---|---|---|
| MSE | 73.15 | 77.91 |
| Cross-Entropy | 74.20 | 79.83 |
| KL Divergence | **75.54** | **82.46** |

Table 20: Hyper-parameters for source training and target adaptation.

| Hyper-parameter | Source Training | Target Adaptation |
|---|---|---|
| Backbone | `ResNet-18` | `ResNet-18` |
| Optimizer | SGD + Nesterov | Adam |
| Momentum | $\{0.1, 0.4, 0.9\}$ | NA |
| Weight Decay | 0.0001 | 0 |
| Learning Rate | $\{0.001, 0.01, 0.02, 0.1\}$ | $\{0.0001, 0.001, 0.002\}$ |
| LR Decay Schedule | Step decay at $\{150, 250, 350\}$ | ReduceLROnPlateau (patience=3) |
| Mini-batch Size | $\{32, 64\}$ | $\{32, 64\}$ |
| Epochs | $\{30, 50, 100\}$ | $\{20, 50\}$ |
| Random Flip | Horizontal/Vertical | Horizontal/Vertical |
| Color Jitter | Brightness/Contrast/Saturation $= 0.5$, Hue $= 0.05$ | Same |
| Image Size | Resize to $225 \times 225$, crop $224 \times 224$ | Same |

# E  HYPERPARAMETER DETAILS

This section summarizes the hyperparameters used for both source pre-training and target adaptation, as presented in Table 20. We report settings for optimizer types, learning rate schedules, and batch sizes. All experiments use a fixed `ResNet-18` backbone to ensure fair comparison across methods. The chosen values follow standard SFDA practices and are selected based on source-domain validation performance.

