# OpenReview forum: "Personalized Feature Translation for Expression Recognition: An Efficient Source-Free Domain Adaptation Method"
_ICLR.cc/2026/Conference — ICLR 2026 Poster_

### Official Review · Reviewer_2fSs · 2025-10-30

**Soundness:** 3
**Presentation:** 2
**Contribution:** 3
**Rating:** 6
**Confidence:** 2

**Summary:**

This paper presents Personalized Feature Translation (PFT), a novel source-free domain adaptation method for facial expression recognition. PFT performs personalized adaptation by translating features in the latent space, requiring only neutral expression data from target subjects. This approach eliminates the need for complex image synthesis and source data access during adaptation, while maintaining high computational efficiency. Extensive experiments demonstrate that PFT consistently outperforms state-of-the-art methods across multiple datasets。

**Strengths:**

The key strength of the proposed PFT method lies in its innovative feature-space translation paradigm. This method avoids the need for complex image synthesis and achieves computational efficiency through lightweight parameter adaptation. It demonstrated superior performance over state-of-the-art methods across four FER benchmarks.

**Weaknesses:**

1 While the empirical results are strong, the paper lacks an analysis explaining the reasons why feature-space translation is more effective.

2 The experimental validation is centered primarily on FER. It would be valuable to discuss the potential of PFT for other tasks that require subject-specific adaptation, such as face recognition or person re-identification.

**Questions:**

NA

---

> ### Author Response · Authors · 2025-11-21
> **Official Comment by Authors**
>
> We thank the Reviewer for the positive and encouraging assessment, and for recognizing the novelty of PFT as a personalized, source-free adaptation method that operates in latent feature space using only neutral target data. We appreciate the Reviewer’s acknowledgment that our approach removes the need for complex image synthesis and source data access, while remaining computationally efficient and outperforming state-of-the-art methods across four FER benchmarks. We address each identified weakness below and will incorporate the Reviewer’s suggestions in the revised manuscript. “W” indicates weakness, “Q” indicates question, while “A” is our answer.
>
> **“W-1:” Justification for why latent-space translation is more robust than image-level translation.**
>
> **“A-1:”** Intuitively, translating in feature space is an easier and more stable optimization problem than image-level translation because the input to our translator is already a compact, discriminative representation. In our setting, translation is applied after the source backbone has filtered out most nuisance factors (background, illumination, pose) and amplified expression-related cues, so the latent features contain far less noise than raw images. The translator then mainly adjusts identity-related components while enforcing expression consistency through a KL loss on the FER classifier outputs, which directly encourages predicted expressions to remain stable during adaptation.
>
> In contrast, image-level methods must optimize the parameters of a generator G that operates in pixel space and implicitly rely on the composition F(G(x)) to preserve expression. This requires modifying the entire image (face texture, lighting, background, etc.), making it a substantially more complex optimization problem than translating already-discriminative features.
> Additionally, we included t-SNE visualizations comparing source-only features, image-level translation, and our feature-space PFT for each class in the revised manuscript (Sec. 5, *Qualitative Analysis via t-SNE Visualization,* Figure 6). These plots show that, after PFT, target features align more tightly with source distributions in latent space, indicating that feature-space translation both reduces domain shift and preserves expression structure more effectively than image-level translation.
>
> **“W-2:” Need discussion of PFT’s applicability to tasks like face recognition or re-ID.**
>
> **“A-2:”**  We focus on video FER because it is a realistic and particularly demanding application for subject-specific adaptation. Indeed, expressions are subtle, identity and expression are strongly entangled, and the availability of only neutral video SFDA is realistic for privacy-sensitive applications (e.g., digital health) where collecting diverse emotional expressions per subject is difficult. Under these conditions, image-level translation easily washes out or hallucinates exactly the finer expressions that matter for FER, whereas PFT’s feature-space translation is better at preserving expression structure.
>
> At the same time, the core ingredients of PFT, (i) a frozen task network, (ii) a lightweight translator on top of its features, and (iii) a prediction-consistency loss combined with a style-alignment loss—are not inherently restricted to FER. In principle, the same framework could be adapted to other tasks that require subject-specific or camera-specific normalization, such as face recognition or person re-identification. In these cases, “content’’ would correspond to identity, while “style’’ would capture nuisance factors such as pose, expression, illumination, or camera characteristics. Translator pre-training could then use same-identity pairs under different conditions, enforcing identity-consistent predictions while aligning statistics to a reference view.
>
> However, exploring these applications would substantially expand the scope of the current work and would require domain-specific design and evaluation. As such, we regard these extensions as promising future directions rather than part of the present study, which is centered on the FER problem.

---

### Official Review · Reviewer_7pCo · 2025-10-30

**Soundness:** 3
**Presentation:** 3
**Contribution:** 2
**Rating:** 4
**Confidence:** 3

**Summary:**

This paper proposes Personalized Feature Translation (PFT), a source-free domain adaptation method for facial expression recognition that operates entirely in the feature space. By translating target subject features toward source-style prototypes using only neutral expressions, PFT achieves higher accuracy with significantly lower computational cost than image-based SFDA methods.

**Strengths:**

This paper introduces a highly original and impactful approach to source-free domain adaptation for facial expression recognition. By formulating a novel feature-level translation method that operates using only neutral target data, it achieves state-of-the-art performance while being dramatically more efficient than image-based alternatives. The work is exceptionally well-supported through rigorous experiments on four diverse benchmarks and presents a practical solution to key real-world constraints like data privacy and computational cost.

**Weaknesses:**

1．	While the proposed PFT method is well-motivated, the paper lacks a clear theoretical or intuitive explanation of why feature translation in latent space is inherently more robust than image-level translation for expression preservation.

2．	The paper lacks a rigorous explanation or analysis of how the proposed losses ensure that the translator modifies only identity-related features while preserving expression-related ones.

3．	The proposed PFT method relies on pre-training a feature-space translator, but there is no analysis of training stability or overfitting risks. Given the challenge of disentangling expression and identity, is there a risk of overfitting to identity features? Are there issues with convergence or conflicting objectives (style vs. expression)?

4．	The experiments are conducted on only 10 target subjects per dataset. This small sample size raises concerns about the statistical significance and generalizability of the results. The authors should include confidence intervals or perform cross-validation across multiple random splits of target subjects.

5．	The paper mentions hyperparameters such as λexpr and λstyle, but does not discuss their sensitivity or how they were tuned.

**Questions:**

review the Weaknesses

---

> ### Author Response · Authors · 2025-11-21
> **Rebuttal by Authors**
>
> We thank the Reviewer for the constructive and very positive assessment, and for recognizing the originality and practical impact of PFT as a source-free, feature-level adaptation method that uses only neutral target data. We appreciate the Reviewer’s acknowledgment of our state-of-the-art performance, substantial computational efficiency compared to image-based approaches, and the practical relevance to privacy- and resource-constrained settings, as demonstrated across four diverse benchmarks. Similar positive feedback on novelty, efficiency, and real-world applicability is echoed by the other Reviewers. We address each identified weakness below and will integrate the Reviewer’s suggestions into the revised manuscript. “W” indicates weakness, “Q” indicates question, while “A” is our answer.
>
> **“W-1:” Justification for why latent-space translation is more robust than image-level translation.**
>
> **“A-1:“**  Intuitively, translating in feature space is an easier and more stable optimization problem than image-level translation because the input to our translator is already a compact, discriminative representation. In our setting, translation is applied after the source backbone has filtered out most nuisance factors (background, illumination, pose) and amplified expression-related cues, so the latent features contain far less noise than raw images. The translator then mainly adjusts identity-related components while enforcing expression consistency through a KL loss on the FER classifier outputs, which directly encourages predicted expressions to remain stable during adaptation.
> In contrast, image-level methods must optimize the parameters of a generator G that operates in pixel space and implicitly rely on the composition F(G(x)) to preserve expression. This requires modifying the entire image (face texture, lighting, background, etc.), making it a substantially more complex optimization problem than translating already-discriminative features.
> Additionally, we included t-SNE visualizations comparing source-only features, image-level translation, and our feature-space PFT for each class in the revised manuscript (Sec. 5, *Qualitative Analysis via t-SNE Visualization,* Figure 6). These plots show that, after PFT, target features align more tightly with source distributions in latent space, indicating that feature-space translation both reduces domain shift and preserves expression structure more effectively than image-level translation.
>
> **“W-2:” Evidence of disentangling style vs. expression in the translator.**
>
> **“A-2:“** To directly assess the specialization, we analyzed the similarity structure of the learned embeddings on the source data (see *Sec. 5, Ablation Studies, Expression and Identity Specialization in Embeddings,* Table 7). For each branch, we computed average cosine similarities between: (1) pairs of samples with the same emotion but different subjects (fixed expr.), and (2) pairs of samples with the same subject but different emotions (Fixed subj.). We observed that in the expression branch, embeddings of different subjects sharing the same emotion are substantially more similar (0.75) than embeddings of the same subject across different emotions (0.40). However, in the identity branch, embeddings of the same subject across different emotions are more similar (0.85) than embeddings of different subjects with the same emotion (0.53). This clear reversal of the similarity pattern indicates that the expression branch is largely invariant to subject identity and primarily encodes expression-related information, while the identity branch captures subject-specific characteristics and is less influenced by expression. These findings empirically support our claim that the proposed training objectives lead to a decoupling of identity and expression in the two branches.
>
> | **Branch**      | **Fixed expr.** | **Fixed subj.** |
> |-----------------|------------------|------------------|
> | **Expression**  | 0.75             | 0.40             |
> | **Identity**    | 0.53             | 0.85             |

---

> ### Author Response · Authors · 2025-11-21
> **Next Response**
>
> **“W-3:” Lack of analysis on translator training stability and overfitting risks.**
>
> **“A-3:“** PFT is designed to mitigate instability and overfitting when disentangling identity and expression. First, we freeze the backbone F and classifier C and only train the small translator T. This limits the capacity of the adapted part and helps keep the global decision boundary of C fixed, reducing the chance that the model learns identity-specific shortcuts. Second, T is always optimized under an expression-consistency loss​: during both source pre-training and target adaptation, we penalize changes in the classifier’s output, via KL loss, which encourages the expression prediction to remain stable after translation. Third, the style term is intentionally weak and structured: we match only spatial statistics (means and variances) from early layers, so identity alignment acts as a mild regularizer rather than a dominant objective.
>
> Our ablations in Sec. 5 (*Ablation Studies, Impact of Expression and Style losses,* Table 6) show that removing either the style or expression term harms performance, suggesting that these losses are complementary rather than conflicting. To address the concern that target features might collapse onto a single source identity, we additionally included in the revision a histogram of nearest source prototypes in Sec. 5 (*Ablation Studies, Target Sample Distribution Across Source Subjects,* Figure 5) for all translated target frames. The distribution is spread over many source subjects (no single subject dominates), indicating that PFT does not overfit by mapping all targets to one identity, but instead learns a smooth, subject-aware alignment across the source population.
>
> **“W-4:” Reviewer raises concerns about statistical reliability and recommends adding confidence intervals or cross-validation.**
>
> **“A-4:“** We agree that evaluating over multiple random subject splits (e.g., via cross-validation) would provide an even stronger assessment of statistical significance and generalizability. In this work, however, we follow the standard personalized domain adaptation protocol used in recent subject-specific FER studies (Zeeshan et al., 2024; 2025; Sharafi et al., 2025), which also adopt 10 fixed target subjects per dataset to enable a fair and direct comparison.
> For BioVid, the 10 target subjects are:
>
>  {071309_w_21, 073109_w_28, 073114_m_25, 080314_w_25, 081014_w_27, 081609_w_40, 091809_w_43, 100909_w_65, 101609_m_36, 112009_w_43},
>
>  which cover both genders and span a wide age range (21–65 years). A similar diversity-oriented selection strategy is applied to StressID, BAH, and Aff-Wild2 whenever metadata is available. Moreover, each target subject contributes hundreds to thousands of frames, so the reported per-subject F1 and accuracy values are computed over large sample sizes.
>
> To further assess robustness, we additionally repeated PFT training on BioVid with three independent random seeds and obtain an average F1 score of 78.43 ± 0.25% (mean ± standard deviation over seeds, computed from the average over the 10 target subjects), indicating that our method is stable across runs. A more extensive cross-validated evaluation over multiple random target splits is an interesting direction, but it is beyond the scope and computational budget of the present submission.
>
> | **Method** | **sub-1**       | **sub-2**       | **sub-3**       | **sub-4**       | **sub-5**       | **sub-6**       | **sub-7**       | **sub-8**       | **sub-9**       | **sub-10**      | **Avg**         |
> |------------|------------------|------------------|------------------|------------------|------------------|------------------|------------------|------------------|------------------|------------------|------------------|
> | **PFT**    | 80.19 ± 0.48     | 71.92 ± 0.15     | 90.13 ± 0.21     | 81.71 ± 0.25     | 92.82 ± 0.67     | 70.37 ± 0.54     | 84.38 ± 0.15     | 79.60 ± 0.44     | 74.66 ± 0.52     | 58.45 ± 0.48     | 78.43 ± 0.25     |
>
> **“W-5:” Lack of sensitivity analysis for λexpr and λstyle.**
>
> **“A-5:“** These coefficients ( ​λexpr and λstyle) control the trade-off between preserving the target expression and aligning the subject identity in the translated features. We tuned both using a small grid search on a held-out validation split and selected λ_expr = 1.0, λ_style = 0.3.
>
> To directly address the reviewer’s concern about hyperparameter sensitivity, we added a dedicated analysis for these coefficients (see *Sec. 5, Ablation Studies: Impact of Expression and Style Losses,* Figure 4). On BioVid, we vary λ_expr and λ_style over a range of non-zero values (λ_expr ∈ {0.1, 0.3, 1.0, 3.0} and λ_style ∈ {0.1, 0.3, 1.0}) and report the resulting target accuracy. The curves exhibit a broad performance plateau around the chosen values, with only small accuracy variations when scaling either coefficient within this range. This indicates that PFT is not overly sensitive to the precise choice of λexpr​ and λstyle​.

---

### Official Review · Reviewer_nYCY · 2025-10-31

**Soundness:** 2
**Presentation:** 2
**Contribution:** 2
**Rating:** 4
**Confidence:** 4

**Summary:**

: This paper proposes a personalized feature translation method named PFT, which efficiently achieves domain adaptation for facial expression recognition models using only target user's neutral expression data. It outperforms existing methods across multiple datasets while significantly reducing computational costs.

**Strengths:**

（1） Structure: The paper features a clear structure, with algorithms presented in easily understandable diagrams. The problem definition is precise, and the motivation is well-articulated.
（2） Innovation: The experiments introduce a personalized translation approach within the feature space, circumventing the instability inherent in traditional image-based methods.
（3） Quality: The experiment design is rigorous, validating the method's effectiveness across four distinct datasets. Detailed ablation studies are provided to demonstrate the contribution of each component.

**Weaknesses:**

（1） Although comparisons are made with multiple SFDA methods, the paper does not include more recently proposed personalized approaches based on generative models or meta-learning.
（2） The paper notes performance degradation on elderly subjects but does not propose adaptive strategies for age differences. Further exploration of stratified or age-aware adaptation methods is recommended.

**Questions:**

（1） Compared to models like SHOT and NRC that also update only partial parameters, how does PFT perform in terms of the number of iterations and total time required to reach convergence?
（2） The core method involves using style loss to align identity features while employing expression loss to preserve facial emotion information. Is there any quantitative evidence demonstrating that the translator network indeed learns decoupled representations of these two factors? If the reference image itself carries strong non-neutral expressions, could this contaminate the translation process, leading to loss or confusion of expression information?
（3） Was comparing PFT against such lighter-weight baselines considered to more clearly demonstrate the added value of feature translation over simple normalization or alignment strategies?

---

> ### Author Response · Authors · 2025-11-21
> **Rebuttal by Authors**
>
> We thank the Reviewer for the constructive and positive assessment, including the clear structure and problem definition, the intuitive algorithmic diagrams, the novelty of personalized feature translation in feature space, and the rigorous experiments with extensive ablations and reduced computational cost. Similar positive feedback on novelty, clarity, and empirical validation is shared by the other reviewers. We address each of the identified weaknesses below and will incorporate the Reviewer’s suggestions in the revised manuscript.  “W” indicates weakness, “Q” indicates question, while “A” is our answer.
>
> **“W-1:Missing Comparison with personalized generative or meta-learning SFDA approaches.”**
>
> **“A-1”:** We would like to clarify that we already include the DSFDA method (Sharafi et al., 2025) in our comparisons, which is precisely a personalized, generative source-free method for FER. DSFDA performs subject-specific adaptation in the image space through a GAN-based translation module that synthesizes and adapts non-neutral target facial images to compensate for missing classes and reduce the source–target domain gap. In our experiments in Sec. 4.2 (Comparison with State-of-The-Art Methods). DSFDA is one of the main baselines, and, as reported in Tables 1–4, it is consistently outperformed by our PFT approach across all source–target settings, despite its heavier image-level generative component. To the best of our knowledge, DSFDA is the only directly comparable personalized generative SFDA method for FER/pain recognition. If the reviewer has suggestions for additional directly comparable personalized generative or meta-learning–based methods, we would be happy to include them in a revised version of the manuscript.
>
> **“W-2: Missing adaptive methods to address performance drop on elderly subjects.”**
>
> **“A-2”:** In our subject-based setting on the BioVid dataset, we indeed observe performance degradation for the elderly subject, indicating that elderly individual faces are particularly challenging in our setup. This is consistent with prior FER and emotion-recognition studies showing that models systematically perform worse on faces captured for older adults, largely because they tend to display less prototypical, lower-intensity expressions and exhibit age-related changes in facial morphology (e.g., wrinkles, reduced muscle tone) that alter the observable action units. For example, Guo et al., 2013; SONMEZ, 2019; Kim et al., 2021a, all report significantly lower recognition accuracy for older subjects compared to younger or middle-aged groups, and a recent systematic review focused on elderly FER echoes these findings and highlights age bias as a persistent challenge in current systems.
> In our PFT framework, we already construct source pairs that account for age, gender, and pose (see *Sec. 5, Ablation Studies, Impact of Source Subject pairing Strategy,* Figure 3), using a landmark-based strategy that aligns facial landmarks via Procrustes analysis and matches subjects with similar head pose under gender and age (≤10 years) constraints. However, the remaining degradation on the subject aged 60+ suggests that these age-related effects are not fully compensated by our current translation strategy. Building on the above literature, a natural extension of PFT is to introduce age-aware pairing and adaptation. Concretely, during translator pretraining we can stratify by age and, for elderly subjects, select reference images from younger subjects that share the same emotion label. This allows the translator to explicitly learn how to bridge age-related appearance differences while preserving the underlying expression. Motivated by the above literature, we are extending PFT with age-aware pairing and adaptation. During translator pretraining, we stratify subjects by age group and, for elderly subjects, select reference samples from younger but expression-consistent subjects to explicitly teach the translator how to bridge age-related appearance differences. Preliminary results show a ~+3% improvement in F1 for the elderly subject, indicating that explicit cross-age reference selection helps mitigate age-related bias. We are completing the full set of evaluations and will report the final quantitative results in the camera-ready version.

---

> ### Author Response · Authors · 2025-11-21
> **Next Response**
>
> **“Q-1: Comparison of PFT’s iteration/time efficiency vs. SHOT and NRC.”**
>
> **“A-1:”** The revised manuscript now includes an explicit convergence analysis in Sec. 4.2 (*Comparison with State-Of-The-Art Methods*, Table 5). We compare accuracy, iteration counts, and wall-clock time under identical settings. Competing approaches require substantially more iterations to reach their peak performance, whereas our method converges in a fraction of the time and achieves markedly higher accuracy. Despite also updating only partial parameters, it delivers superior performance with 5–9× fewer iterations and significantly lower runtime than existing SFDA baselines.
>
>
> | **Method**                    | **Accuracy (%)** | **Iterations** | **Time (s)** |
> |------------------------------|-----------------|---------------|-------------|
> | SHOT (Liang et al., 2020)    | 50.35           | 1155          | 54.0        |
> | SFDA-DE (Liang et al., 2020) | 62.88           | 1400          | 65.5        |
> | NRC (Yang et al., 2021)      | 60.31           | 705           | 75.0        |
> | **PFT (Ours)**               | **82.46**       | **135**       | **0.95**    |
>
>
> **“Q-2: Evidence of disentangling style vs. expression in the translator.”**
>
> **“A-2:”** To directly assess the specialization, we analyzed the similarity structure of the learned embeddings on the source data (see *Sec. 5, Ablation Studies, Expression and Identity Specialization in Embeddings,* Table 7). For each branch, we computed average cosine similarities between: (1) pairs of samples with the same emotion but different subjects (fixed expr.), and (2) pairs of samples with the same subject but different emotions (Fixed subj.). We observed that in the expression branch, embeddings of different subjects sharing the same emotion are substantially more similar (0.75) than embeddings of the same subject across different emotions (0.40). However, in the identity branch, embeddings of the same subject across different emotions are more similar (0.85) than embeddings of different subjects with the same emotion (0.53). This clear reversal of the similarity pattern indicates that the expression branch is largely invariant to subject identity and primarily encodes expression-related information, while the identity branch captures subject-specific characteristics and is less influenced by expression. These findings empirically support our claim that the proposed training objectives lead to a decoupling of identity and expression in the two branches.
>
> | **Branch**      | **Fixed expr.** | **Fixed subj.** |
> |-----------------|------------------|------------------|
> | **Expression**  | 0.75             | 0.40             |
> | **Identity**    | 0.53             | 0.85             |
>
>
> **“Q-3: Comparison against lighter-weight normalization/alignment baselines.”**
>
> **“A-3:”** We agree that comparing PFT to lighter-weight normalization/alignment strategies is important to isolate the added value of feature translation. In the revised manuscript, we make this explicit by comparing PFT with SFDA-DE (Liang et al., 2020), a source-free method that adapts only the classifier and normalization statistics on the target domain without image synthesis or generative modeling.
>
> We implemented SFDA-DE in our subject-specific BioVid setting using the same backbone, data splits, and training protocol as PFT. The subject-wise F1-scores shown below indicate that PFT consistently outperforms SFDA-DE across all target subjects with a higher average F1-score. Since both methods have comparable capacity and adaptation budget, this gap cannot be attributed to simple normalization or classifier tuning; it reflects the benefit of modeling subject-specific feature translation in latent space.
>
> To further support this analysis, Sec. 4.2 (*Comparison With State-of-the-Art Methods*, Table 5) reports accuracy, iterations, and adaptation time under identical hardware. These results confirm that PFT remains lightweight while outperforming normalization- or alignment-based SFDA approaches. Overall, the improvement stems from feature translation rather than increased adaptation capacity.
>
>
> | **Method** | **sub-1** | **sub-2** | **sub-3** | **sub-4** | **sub-5** | **sub-6** | **sub-7** | **sub-8** | **sub-9** | **sub-10** | **Avg** |
> |-----------|-----------|-----------|-----------|-----------|-----------|-----------|-----------|-----------|-----------|------------|--------|
> | **SFDA-DE** | 77.70 | 54.36 | 56.26 | 53.96 | 77.50 | 59.30 | 48.16 | 75.96 | 72.71 | 52.90 | 62.88 |
> | **PFT**     | **80.65** | **71.75** | **90.26** | **81.54** | **92.68** | **70.06** | **84.26** | **79.29** | **74.53** | **58.08** | **78.31** |

---

> > ### Comment · Reviewer_nYCY · 2025-11-25
> >
> > Authors have solved my questions, I will rise my score to 6.

---

### Author Response · Authors · 2025-11-21
**Official Comment by Authors**

We would like to express our sincere gratitude to all the reviewers for their thoughtful and constructive feedback. Their comments have greatly helped us clarify and strengthen the paper. We appreciate the positive recognition of our work, including the novelty of Personalized Feature Translation (PFT) as a source-free, feature-level adaptation method (2fSs), the practicality of our neutral-only target setting (nYCY, 7pCo), the clarity of the presentation (nYCY), and the strong empirical results and computational efficiency (2fSs). The strong experimental validation across four challenging FER benchmarks is emphasized by nYCY and 7pCo also .

Reviewers mainly asked us to further clarify why feature-space translation is more robust than image-level translation, how the proposed losses lead to a disentanglement of identity and expression, and how stable and lightweight PFT is compared to SFDA baselines such as SHOT, NRC, and normalization/alignment-based methods. We have addressed all comments point by point, integrated the requested clarifications and analyses into the manuscript, and added new results and plots.

---

### Author Response · Authors · 2025-12-04
**Final message to Area Chair**

Dear Area Chair,

Thank you for handling our submission and coordinating the insightful reviews. For your convenience, we briefly summarize: (i) our paper, (ii) the main comments and questions from reviewers with high-level responses, and (iii) the key changes made to the manuscript (marked in red in the revised version).

**Summary of the paper:**

Our paper introduces Personalized Feature Translation (PFT), a source-free domain adaptation (SFDA) method for subject-specific adaptation of facial expression recognition (FER) systems. It assumes access to a source-trained FER model and a small neutral-only video from each target subject, but no source data nor non-neutral target expressions. We introduce a lightweight translator network that operates on top of frozen backbone, translating target features toward source-style prototypes while preserving expression predictions. The translator is pre-trained on source subjects with a combination of expression-consistency loss and style (identity) alignment loss, which encourages the extraction of expression and identity using two branches. During subject-specific adaptation, only the translator is updated on neutral target data. Using four challenging FER datasets (including pain, stress, ambivalence/hesitancy, and basic emotions), PFT consistently outperforms both image-level generative methods (e.g., DSFDA) and image-translation SFDA baselines, considerably more efficient. For example, on the BioVid dataset PFT reaches 82.46% accuracy in 135 iterations (~0.95 s), whereas SHOT, SFDA-DE, and NRC reach 50.35%, 62.88%, and 60.31% with 1155, 1400, and 705 iterations (54.0–75.0 s), respectively.

**Paper’s strengths:**

We truly appreciate the following positive recognition of our work shared by reviewers.
- The novelty and impact of PFT as a subject-specific, and feature translation method that relies only neutral target video data (7pCo, nYCY, 2fSs);
- The practical relevance of the neutral-only and source-free setting for privacy- and resource-constrained cases, especially in digital health applications (7pCo, 2fSs);
- The clear structure, intuitive diagrams, and well-motivated problem formulation and presentation (nYCY, 7pCo);
- The rigorous experiments and ablations on four challenging FER benchmarks (nYCY, 7pCo, 2fSs);
- State-of-the-art performance of PFT combined with its substantially lower computational complexity when compared with image-based and strong feature-based SFDA baselines (7pCo, 2fSs).

**Reviewers’ suggestions to improve our manuscript:**

- We included new t-SNE plots (Sec. 5, Fig. 6) to clarify why PFT is more effective than image-based translation, showing that PFT yields greater alignment of target features to source class distributions while preserving class structure (7pCo, 2fSs).
- We included a cosine-similarity analysis (Sec. 5, Table 7) to provide more quantitative evidence and clarification of disentanglement, revealing reversed similarity patterns in the expression vs. identity branches (nYCY, 7pCo).
- We included additional ablations and a prototype histogram (Sec. 5, Table 6 & Fig. 5) to clarify training stability and the effect of the joint style and expression objectives, showing stable training, complementary losses, and no identity collapse (7pCo).
- We included a convergence table with SHOT, SFDA-DE, and NRC (Sec. 4.2, Table 5) to provide a clearer comparison of efficiency and complexity, showing that PFT achieves higher accuracy with far fewer iterations and lower adaptation time (nYCY).
- We included a sensitivity analysis (Sec. 5, Fig. 4) to clarify the impact of λexpr​ and λstyle, showing that performance remains stable over a broad range of parameter values (7pCo).

In our rebuttal below, we provided a detailed response to each question raised by reviewers and described the modifications/inclusions to improve the quality of the manuscript. We have indicated the respective changes to the original submitted manuscript.

**Manuscript changes (in red):**

We have integrated several clarifications suggested by reviewers.
- Expanded Sec. 4.2 (*Comparison with State-of-the-Art Methods*) with an efficiency table (Table 5) reporting accuracy, number of iterations, and wall-clock time for PFT and baseline methods.
- Extended Sec. 5 (*Ablation Studies*) with:
  - a similarity analysis of expression vs. identity branches (Table 7) to quantify disentanglement,
  - a sensitivity analysis of the weights λ_expr and λ_style for expression and style losses (Table 6 and Fig. 4),
  - a histogram of nearest source prototypes for translated targets (Fig. 5) to rule out identity collapse.
- Added t-SNE visualizations in Sec. 5 (*Ablation Studies*) comparing source-only vs PFT and source-only vs image-based translation (Fig. 6).
- Added experimental results with additional (CNN and ViT) backbones in the supplementary material (Sec. D.4), showing that PFT gains are consistent beyond the main backbone used in the original manuscript.

---

### Meta-Review · Area_Chair_6Gjp · 2026-01-11

**Summary:**

This paper proposes Personalized Feature Translation (PFT) for source-free domain adaptation in video facial expression recognition under a very constrained setting: no source data and target data contains only neutral expressions. The key idea is to translate in feature space with a lightweight translator (backbone/classifier frozen), pre-trained on source subjects and then adapted per target subject using neutral-only videos. Across four FER benchmarks, PFT reports consistent gains over SFDA baselines and faster adaptation. Reviewer opinions are mixed.

**Reviewer Concerns:**

**Addressed by rebuttal:**
- Efficiency vs partial-update baselines (nYCY): Added convergence/time table; PFT reaches higher accuracy with far fewer iterations and wall-clock time.
- “Does it really disentangle identity vs expression?” (nYCY, 7pCo): Added cosine-similarity analysis showing the expected reversal across expression/identity branches; supports specialization.
- Why feature-space translation is more robust than image translation (7pCo, 2fSs): Provided intuitive argument + added t-SNE plots showing better class-structure preservation and alignment vs image-based translation.
- Stability / overfitting / identity collapse risk (7pCo): Added ablations and prototype histogram indicating no collapse to one identity; argues complementary objectives.
- Hyperparameter sensitivity (7pCo): Added sensitivity analysis.
- “Need lighter alignment baselines” (nYCY): They compare to SFDA-DE and show large gap with similar adaptation budget.

**Remaining concerns:**
- Generalization/statistical rigor (7pCo): They keep the “10 fixed subjects” protocol for comparability; add 3-seed stability, but no full cross-val / multiple splits / confidence intervals beyond seeds.
- Coverage of “recent personalized/meta-learning” methods (nYCY): They claim DSFDA already covers personalized generative SFDA; if there exist other close FER SFDA personalization/meta-learning methods, they are not added.
- Age-drop handling (nYCY): They provide a plausible age-aware extension with preliminary +3% F1 on elderly subject, but not fully validated in the main results yet.
- Scope beyond FER (2fSs): Addressed as future work; fine.

**Reviewer Scores:**

- nYCY: 4, conf 4 → explicitly said they would raise to 6 after rebuttal.
- 7pCo: 4, conf 3 (no explicit post-score update shown).
- 2fSs: 6, conf 2.
- There is a “request to reduce review load” from reviewer 31wi, but no official review from them is present.

---

### Decision · Program_Chairs · 2026-01-26

Accept (Poster)